# WebWorld: A Large-Scale World Model for Web Agent Training

Zikai Xiao [† 1]   Jianhong Tu [2]   Chuhang Zou [3]   Yuxin Zuo [2]   Zhi Li [2]   Peng Wang [2]   Bowen Yu [2]   Fei Huang [2]
Junyang Lin [* 2]   Zuozhu Liu [* 1]

## Abstract

Web agents require massive trajectories to generalize, yet real-world training is constrained by network latency, rate limits, and safety risks. We introduce **WebWorld** series, the first open-web simulator trained at scale. While existing simulators are restricted to closed environments with thousands of trajectories, WebWorld leverages a scalable data pipeline to train on 1M+ open-web interactions, supporting reasoning, multi-format data, and long-horizon simulations of 30+ steps. For intrinsic evaluation, we introduce WebWorld-Bench with dual metrics spanning nine dimensions, where WebWorld achieves simulation performance comparable to Gemini-3-Pro. For extrinsic evaluation, Qwen3-14B trained on WebWorld-synthesized trajectories improves by +9.2% on WebArena, reaching performance comparable to GPT-4o. WebWorld enables effective inference-time search, outperforming GPT-5 as a world model. Beyond web simulation, Web-World exhibits cross-domain generalization to code, GUI, and game environments, providing a replicable recipe for world-model construction. Code and models are publicly available.

## 1. Introduction

Autonomous web agents based on large language models (LLMs) are widely used for various web tasks, as they can leverage strong language priors to reason and plan. However, their ability to reliably execute actions in real-world browser environments remains limited. In the experience era (Silver & Sutton, 2025), continuous interaction with the environment is key to building more robust and action-

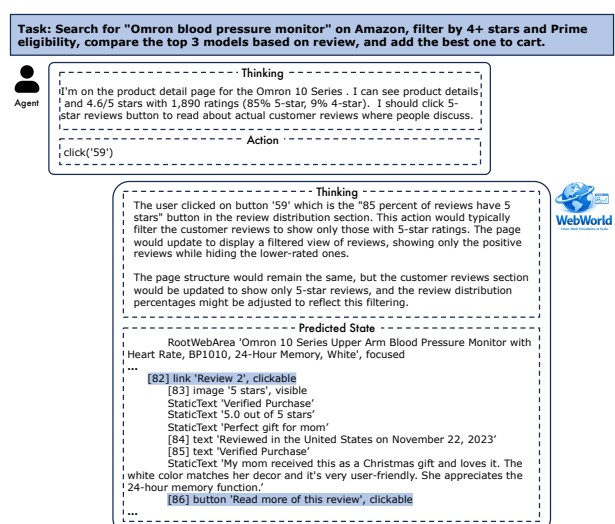

*Figure 1.* WebWorld Example. Left: Agent. Right: WebWorld. Full trajectories are provided in the **supplementary materials.**

oriented agents (Yang et al., 2025; Xi et al., 2025; Huang et al., 2025; Hui et al., 2024). Nevertheless, scaling web agents for large-scale real-world interactions remains difficult. Collecting trajectories is slow due to network latency, page loading times, and rate limits, and many websites employ anti-crawling or access restrictions. Moreover, interactions require careful safety considerations, as some actions (e.g., submitting sensitive forms or initiating transactions) may be irreversible (Ram, 2025; Bonagiri et al.). Therefore, web world models provide a potential solution by enabling agents to train in simulated environments (Anonymous, 2025a; Feng et al., 2025; Song et al., 2026).

Recent work demonstrates the effectiveness of LLM-based world models to produce large quantities of synthetic trajectories, substantially improving agent learning (Team et al., 2025; DeepSeek-AI, 2024). In web scenarios, while prompting proprietary frontier LLMs as a world model (Wang et al., 2025; Li et al., 2025a) has shown initial promise for agent training, more recent efforts have focused on training a web world model (Chae et al., 2025; Chen et al., 2025; Gao et al., 2025). However, existing models exhibit poor generalization because the data pipeline is not easily scalable. First, they rely on a narrow set of agent tasks, resulting in datasets restricted to 1k–10k trajectories. Furthermore,

[†]Work done during internship at Qwen Team. [*]Co-corresponding authors. [1]Zhejiang University, Hangzhou, China [2]Qwen Team, Alibaba Group, Hangzhou, China [3]Independent Researcher. Correspondence to: Junyang Lin <junyang.ljy@alibaba-inc.com>, Zuozhu Liu <zuozhuliu@intl.zju.edu.cn>.

*Proceedings of the 43rd International Conference on Machine Learning*, Seoul, South Korea. PMLR 306, 2026. Copyright 2026 by the author(s).

*Table 1.* **Comparison of World Models for Web Agents.** We categorize existing approaches into **API-based prompting methods** and **trained world models**. Unlike proprietary API-based simulators or prior open-weights models restricted to closed web environments, **WebWorld** is a generalist world model trained on large-scale (1M+) real-world trajectories, supporting **internal reasoning**, **long-horizon** consistency, and **open-web** generalization.

| Model | Size | Open Access | | Data Source | | Formats | Model Capabilities | | |
|---|---|---|---|---|---|---|---|---|---|
| | | Model | Data | Type | Scale | | Open-Web | Reason | Long-Horizon |
| *API-Based Prompting Methods* | | | | | | | | | |
| UI-Simulator (Wang et al., 2025) | GPT-4o-mini | - | - | Prompting | - | A11y | ✓ | ✓ | ✓ |
| Simia (Li et al., 2025a) | o4-mini | - | - | Prompting | - | Text | ✓ | ✓ | ✓ |
| *Trained World Models* | | | | | | | | | |
| DreamGym (Chen et al., 2025) | 8B | × | × | Benchmarks | ~14K | Text | × | ✓ | Full Traj. |
| WebEvolver (Fang et al., 2025) | 70B | × | × | Self-Gen | ~5K | A11y | × | × | Single-step |
| WMA (Chae et al., 2025) | 8B | ✓ | ✓ | Benchmarks | ~14K | Text | × | ✓ | Single-step |
| Word2World (Li et al., 2025b) | 8B | ✓ | × | Benchmarks | ~70k | Text | × | × | Full Traj. |
| WebSynthesis (Gao et al., 2025) | 7B | × | ✓ | Benchmarks | ~4K | A11y | × | × | Single-step |
| **WebWorld (Ours)** | **8B/14B/32B** | ✓ | ✓ | **Real-world** | **1.06M** | **Multi\*** | ✓ | ✓ | ✓ |

**Open-Web**: Generalizes to diverse real-world websites beyond limited benchmarks (e.g., WebArena). **Reason**: CoT to explain state transitions before prediction. **Long-Horizon**: Supports long-horizon interaction history (up to 30 turns) for consistent simulation. **Full Traj.**: Uses complete trajectory history. **Single-step**: Only uses the most recent state. **Multi\* Formats**: Supports Text, A11y, HTML, XML, and Markdown. **Self-Gen**: Generated by an agent exploring live websites based on benchmark queries. *Note*: Word2World utilizes a simplified, flattened text for state representation, distinct from A11y Tree.

because data is collected from sandboxes or closed environments for benchmarking purposes, the resulting trajectories lack diversity. These limitations often confine models to single-turn predictions and restricted input formats, while precluding explicit reasoning capabilities, see Table 1.

**We introduce WebWorld ( Figure 1), a large-scale open-web world model series (8B, 14B, and 32B) trained on 1M+ real-world trajectories** (100× more than prior work) that supports reasoning, long-horizon simulation (30+ turns), and multiple input formats (A11y Tree, HTML, etc.). To ensure generalization, we build a scalable, hierarchical data pipeline that greatly expands coverage over prior work.

The data pipeline ( Figure 2.c) first uses rule-based crawlers on websites from pre-training corpora to scalably collect massive amounts of trajectories aligned with the model's pre-training prior (43.3% of total data). Then, agents autonomously explore diverse websites by generating their own tasks, producing large-scale natural interaction data (20.4%). Finally, agents execute predefined tasks to collect task-oriented trajectories (16.1%). This pipeline collects 1M trajectories that inject knowledge into the model. Since real-world trajectories rarely include explicit reasoning, we further fine-tune on 1K CoT samples (0.09%) synthesized by Claude-Opus-4.1 to inject causal reasoning patterns. Experiments validate that the knowledge-then-reasoning-pattern injection recipe is essential for world models ( Table 7).

To holistically assess WebWorld, we introduce **WebWorld-Bench**, an intrinsic benchmark that evaluates models us-

ing Factuality and Web Turing Scores across nine dimensions, from long-horizon simulation to multi-format robustness. WebWorld achieves performance on par with Claude-Opus-4.1 and Gemini-3-Pro, maintaining consistently high scores across all metrics.

Furthermore, we validate WebWorld's utility through two extrinsic scenarios. First, we synthesize 8,000 diverse trajectories using WebWorld with an Abstract-and-Instantiate strategy. Fine-tuning Qwen3-8B on these trajectories achieves +9.9% gains on MiniWob++ and +10.9% on WebArena, with the fine-tuned 14B model reaching performance comparable to GPT-4o. Second, for inference-time lookahead search, we use WebWorld to simulate the next state for action selection, and it outperforms GPT-5 as a world model. We also observe that WebWorld adheres to predictable scaling laws, with performance consistently improving across model sizes without saturation. Finally, beyond web simulation, WebWorld exhibits strong cross-domain generalization to code, GUI, and game environments, validating the web as a general foundation for other world model adaptation.

Our contributions are three-fold: (1) We propose WebWorld, the large-scale web simulator trained on 1M+ real-world trajectories with a scalable hierarchical data pipeline. (2) We introduce WebWorld-Bench, a comprehensive evaluation framework with dual metrics across nine dimensions. (3) We demonstrate that agents trained on WebWorld-synthesized data achieve significant performance gains.

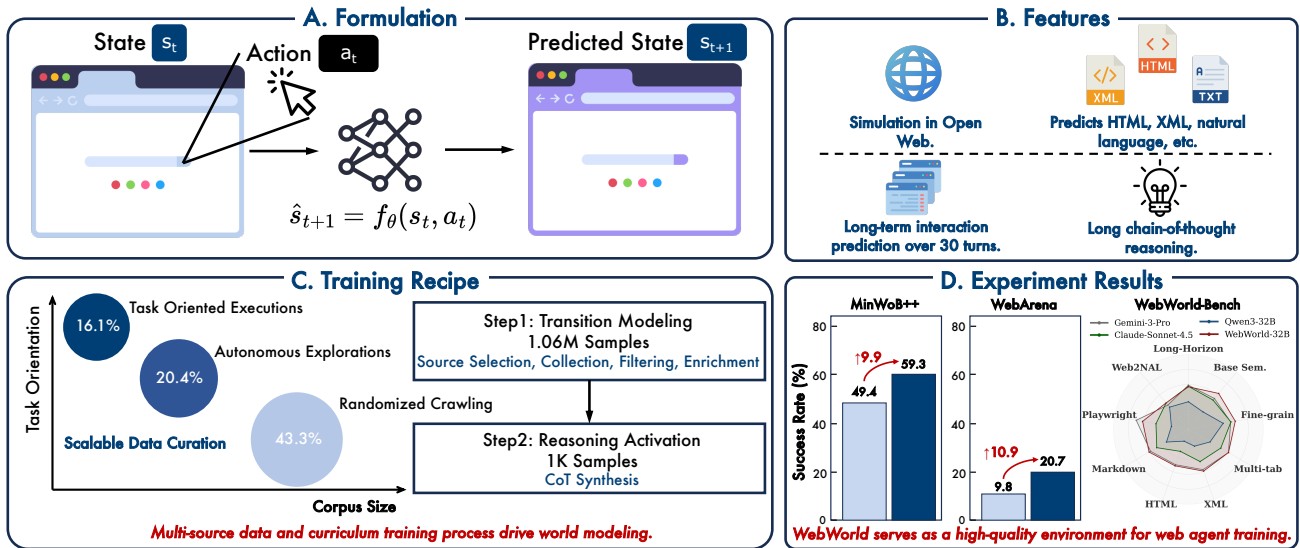

*Figure 2.* Overview of WebWorld. WebWorld is a large-scale world model for the open web, trained on over 1M real-world trajectories. It supports long-horizon, multi-format simulation, enabling agents trained with its data to achieve significant performance gains.

## 2. Related Work

**Web world models** have been explored as a potential approach for web agent training. Early work focused on API-based models utilizing zero-shot or few-shot prompting. For instance, UI-Simulator (Wang et al., 2025) uses Retrieval-Augmented Simulation, employing world models to systematically synthesize trajectories in a controlled manner, specifically targeting the agent's weaknesses for its training. Simia (Li et al., 2025a) generates trajectories from tool specifications, enhancing both offline data synthesis and online reinforcement learning. These approaches highlight the potential of world models for agent training.

Recent research has shifted from prompting to training world models in closed environments. DreamGym (Chen et al., 2025) uses offline trajectories from WebArena and WebShop to train models through experience replay and retrieval-augmented generation (RAG). More advanced agent-driven synthesis methods, including those from Li et al. (2025b) and Gao et al. (2025), employ Monte Carlo Tree Search (MCTS) to explore. WMA (Chae et al., 2025) leverages synthetic tasks and agent exploration to collect trajectories, while WebEvolver (Fang et al., 2025) fine-tunes both world models and agents alternately using co-evolution, where agent-collected data further enhances the models.

While these works still predominantly rely on closed, benchmark environments for data collection, WebWorld targets the open web to improve generalization. By employing a scalable hierarchical collection strategy, WebWorld efficiently captures diverse real-world dynamics.

## 3. Training WebWorld

### 3.1. Overview

We model the browser world as an autoregressive simulator. Given an instruction $I$ and history $h_t = (s_0, a_0, \ldots, s_t, a_t)$ of states and actions, it predicts the next state:

$$s_{t+1} \sim P_\theta(\cdot \mid I, h_t). \tag{1}$$

We instantiate $P_\theta$ with a causal LLM and train it by maximum likelihood on trajectories $\tau = (I, s_0, a_0, \ldots, s_T)$:

$$\mathcal{L}(\theta) = -\mathbb{E}_{\tau \sim \mathcal{D}} \sum_{t=0}^{T-1} \log P_\theta(s_{t+1} \mid I, h_t). \tag{2}$$

To ensure the model generalizes, we align the data source with the pre-training corpus. Specifically, we extract target URLs directly from the metadata of large-scale pre-training corpora and employ a scalable hierarchical collection strategy to harvest trajectories (**Section 3.2**). The collected data is filtered using rule-based checks and LLM-based verification to ensure quality (**Section 3.3**). Subsequently, we apply data augmentation (**Section 3.4**) to enable multi-format prediction. Finally, we adopt a two-stage training curriculum: after initial large-scale dynamics training, we continue fine-tuning with CoT data (**Section 3.5**) to explicitly activate the model's reasoning capabilities.

### 3.2. Data Construction Pipeline

**Data Format** We adopt the A11y Tree as our primary state representation due to its universal applicability across web and GUI environments, high information density, and LLM-friendly structure (Zhou et al., 2023). We extract A11y Trees

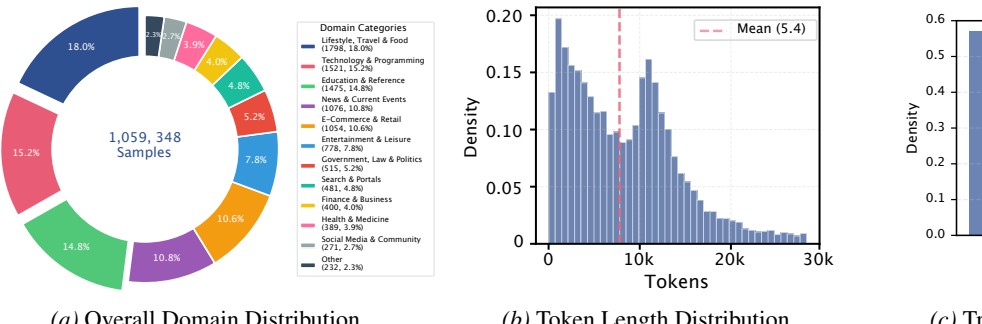

*(a)* Overall Domain Distribution      *(b)* Token Length Distribution      *(c)* Trajectory Turns Distribution

*Figure 3.* **Statistics of the WebWorld Dataset. (a)** Diverse coverage across domains like Lifestyle, Tech, and Education. **(b)** Token distribution showing the model's exposure to varying context lengths. **(c)** Interaction turns distribution confirming the inclusion of long-horizon tasks (up to 30+ steps).

using the Playwright API from BrowserGym (de Chezelles et al., 2025). To prevent overfitting to a single format, we augment training data by converting trajectories into multiple web representations via post-hoc conversion and natural language page descriptions via LLM. Details of data formats can be found in Appendix G.

**Data Source.** *For URL sources,* We primarily extract target URLs from large-scale pre-training corpora—FineWeb (Penedo et al., 2024) (English, ∼618k URLs) and a quality-filtered subset of CCI 3.0 (Wang et al., 2024) (Chinese, ∼64k URLs)—ensuring alignment between the world model's training distribution and the base LLM's pretraining priors. We added curated lists of high-traffic English and Chinese websites (e.g., e-commerce, social media, news portals). *For task queries:* For Level 3 (Task-Oriented Execution), we synthesize task-oriented queries by sampling seed tasks from the Mind2Web training set (Deng et al., 2023) and generating diverse variants via LLM. *Auxiliary data:* We incorporate open-source agent trajectories (Xu et al., 2025) by reformatting trajectories into $(s_t, a_t, s_{t+1})$ tuples, and mix in general instruction-following data (Ding et al., 2023) to preserve conversational abilities.

**Scalable Hierarchical Data Collection.** Existing methods rely on task-directed execution, which ensures relevance but limits scalability. We propose a **scalable**, three-level (hierarchical) pipeline (Figure 2.c) that maintains task relevance through: randomized exploration (*breadth*), autonomous discovery (*realism*), and task synthesis (*task-alignment*).

**Level 1: Randomized Crawling.** To scalably harvest large-scale interaction data, we deploy randomized crawlers on websites extracted from pre-training corpora (FineWeb, CCI 3.0). The crawlers randomly sample executable actions from the current page's A11y Tree—such as clicking buttons, filling forms, or selecting dropdowns—and execute 3-10 step trajectories per website. This ensures the training distribution aligns with the model's linguistic priors from pre-training, maximizing the activation of its innate web

understanding. This stage yields 293K diverse trajectories spanning the structural breadth of the open web.

**Level 2: Autonomous Exploration.** To capture realistic agent–environment interaction dynamics, we deploy LLM-based agents that autonomously explore websites by generating their own exploratory objectives. **We steer trajectory patterns through prompt design.** Prompts encode targeted behavioral priors that induce the interaction patterns desired for world-model learning (Appendix N.4). Concretely, we implement four complementary exploration strategies: *(i) Self-proposed Task*—the prompt instructs the agent to infer a concrete user intent from the current page and execute it; *(ii) Long-horizon dependency*—the prompt forces the agent to produce trajectories where later states depend on earlier actions; *(iii) Composite Action interaction*—the prompt requires multi-action (type/select/click) to avoid trivial navigation-only behavior; *(iv) Curiosity discovery*—the prompt encourages systematic coverage of major sections and features to maximize breadth. Each trajectory spans up to 30 steps, and agents naturally terminate upon exhausting discoverable content or hitting step limits. This stage produces 38K long-horizon trajectories that reflect realistic agent behaviors.

**Level 3: Task-Oriented Execution.** To ensure the model masters task-oriented dynamics, we synthesize explicit web tasks through a three-stage generation pipeline: *(i) Seed extraction*—an LLM analyzes a website and proposes feasible user intents (e.g., "book a flight"); *(ii) Task diversification*—for each seed, we generate multiple task variants by perturbing parameters while maintaining executability on the same website; *(iii) Paraphrase*—we generate semantically similar but linguistically diverse phrasings of each task. Agents then execute these tasks on the corresponding websites, and we retain only successful trajectories. This yields 94K high-quality execution traces where every action is purposeful and goal-directed, capturing the state transitions essential for complex agentic workflows.

Across all levels, we represent page states using A11y Tree, which provides a structured, LLM-friendly abstraction of interactable elements while filtering out rendering noise. The final dataset combines these levels with enriched data (subsection 3.4), totaling 1.06M trajectories ( Table 11).

### 3.3. Filtering

To ensure high data quality and safety, we implement a rigorous dual-stage filtering pipeline applied to both the source URLs and the collected trajectories. We first employ script-based heuristics to verify website reachability and filter out content containing banned keywords (e.g., pornography, gambling, violence). The initial filtering for website reachability leaves 15.7% of the original URLs, of which 85.2% subsequently pass the keyword check. Subsequently, we utilize an LLM to score the remaining sites across four dimensions: *accessibility*, *content suitability*, *interactivity*, and *engineering quality*. Sites scoring below the average or triggering safety violations are discarded. The details of LLM-based URL filtering are illustrated in Figure 6. For the collected trajectory, we apply keyword filtering to eliminate unsafe content. We further prune transitions where an action results in no observable state change (e.g., due to network latency or page loading failures) and discard trajectories exceeding 30k tokens or 30 turns. To avoid introducing the inductive bias of a specific model, we rely exclusively on rule-based trajectory filtering and do *not* employ LLMs for judgment at this stage.

### 3.4. Data Enrichment

Although our collected trajectories provide rich multi-page interactions in A11y Tree format, relying solely on this representation limits model versatility and risks catastrophic forgetting. To address this, we construct a multi-dimensional instruction tuning dataset covering five paradigms, as detailed in Table 2.

In the Web Domain, we implement the Multi-Format Simulator by transpiling trajectories into other formats (Appendix G). We further synthesize Web Generation data (mapping user queries to web pages) and Descriptive Simulation data (converting state changes into textual summaries). In the General Domain, we reformat general QA data into world model prediction tasks. Finally, we mix in general chat data to prevent catastrophic forgetting.

### 3.5. CoT Synthesis

To activate explicit reasoning capabilities, we randomly sample transitions from the 1.06M corpus and prompt Claude-Opus-4.1 to synthesize CoT rationales. Given $(I, s_t, a_t)$, the model generates intermediate reasoning steps—analyzing page structure, interpreting user intent,

*Table 2.* **Data Enrichment Tasks.** Overview of the five auxiliary tasks across Web and General domains. **Notation:** $\mathcal{S}$ represents structured web states (A11y, HTML, XML, Markdown), $\mathcal{T}$ denotes natural language text, and $\mathcal{A}$ indicates agent actions.

| Domain | Task | Input | Output | Description |
|---|---|---|---|---|
| Web | 1. Multi-Format Simulator | $\mathcal{S}_t + \mathcal{A}_t$ | $\mathcal{S}_{t+1}$ | Predict next state in A11y, HTML, XML, or Markdown. |
| | 2. Web Generation | $\mathcal{T}_{intent}$ | $\mathcal{S}_{page}$ | Generate a full webpage structure from user requirements. |
| | 3. Descriptive Simulator | $\mathcal{S}_t + \mathcal{A}_t$ | $\mathcal{T}_{desc}$ | Interpret visual changes and output a text summary. |
| General | 4. General World Model | $\mathcal{T}_t + \mathcal{A}_t$ | $\mathcal{T}_{t+1}$ | Simulate state transitions purely in natural language. |
| | 5. General Chat | $\mathcal{T}_{query}$ | $\mathcal{T}_{response}$ | Standard dialogue to preserve conversational capabilities. |

predicting changes—followed by the next state $s_{t+1}$. We adopt a two-stage curriculum: Stage 1 trains on the full dataset to learn web dynamics; Stage 2 continues training with a small amount of CoT-augmented data to externalize reasoning patterns. As shown in Table 7, our robust pretrained dynamics enable effective reasoning activation with only 1,000 samples, achieving performance that surpasses the base model trained on 10x more CoT data.

### 3.6. Dataset Statistics

We present the dataset statistics in Figure 3. The domain distribution demonstrates balanced coverage across diverse categories, with detailed source breakdowns provided in Figure 5. Furthermore, the dataset exhibits significant variance in complexity, featuring context lengths up to 30k tokens and long-horizon trajectories reaching 30 turns, ensuring the model generalizes effectively to both simple and extended web interactions.

## 4. Benchmarking Web World Model

Existing intrinsic evaluation metrics for world models fall into two categories. *Structural metrics* (Fang et al., 2025) measure DOM tree similarity and element-level alignment, while *semantic metrics* (Chae et al., 2025) use information coverage (ROUGE/BERTScore) between predicted and actual state change descriptions. ViMo (Anonymous, 2025b) extends this with visual similarity and functional availability for mobile GUIs. However, these approaches struggle with open-ended web tasks: structural metrics produce uniformly low scores due to HTML's high variance, while semantic metrics fail to differentiate model capabilities when state changes are complex (Appendix J for detailed analysis).

To address this, **we construct WebWorld-Bench** with two complementary metrics: *Factuality Score* employs pointwise evaluation, where an LLM judge scores whether the predicted state correctly reflects the functional effect of the action, capturing factual correctness on a continuous scale; *Web Turing Score* uses pairwise comparison, where the judge attempts to distinguish simulated states from real ones, assessing perceptual realism through adversarial discrimination. Together, these metrics provide both objective verification and subjective plausibility assessment. We

*Table 3.* **Performance comparison across nine evaluation dimensions.** Each dimension reports both Factuality Score (Fact.) and Web Turing Score (Tur.) in paired columns. Models are categorized into Proprietary and Open-source. The best results in each metric are **bolded**. All scores are normalized to [0, 1], with higher values indicating better performance.

| Model | Long-Horizon (Consistency) | | Base Sem. (Semantics) | | Fine-grain (Sensitivity) | | Multi-tab (Multi-page) | | Multi-format Robustness | | | | | | | | Web2NAL (Nat. Lang.) | | Average (All) | |
|---|---|---|---|---|---|---|---|---|---|---|---|---|---|---|---|---|---|---|---|---|
| | | | | | | | | | XML | | HTML | | Markdown | | Playwright | | | | | |
| | Fact. | Tur. | Fact. | Tur. | Fact. | Tur. | Fact. | Tur. | Fact. | Tur. | Fact. | Tur. | Fact. | Tur. | Fact. | Tur. | Fact. | Tur. | Fact. | Tur. |
| *Proprietary LLMs* | | | | | | | | | | | | | | | | | | | | |
| GPT-4o | 69.2 | 25.0 | 55.3 | 43.0 | 81.0 | 30.0 | 69.9 | 32.0 | 63.5 | 25.0 | 47.3 | 22.0 | 64.1 | 36.0 | 51.3 | 44.0 | 34.2 | 62.0 | 59.5 | 35.4 |
| Claude-Sonnet-4.5 | 78.1 | 37.0 | 60.4 | 42.0 | **81.1** | 34.0 | 68.8 | 31.0 | 60.4 | 31.0 | 42.4 | 20.0 | 63.3 | 34.0 | 51.7 | 36.0 | 31.4 | 61.0 | 59.7 | 36.2 |
| Claude-Opus-4.1 | **82.9** | 34.0 | **72.1** | **53.0** | 79.4 | **35.0** | 79.3 | **43.0** | **76.4** | **42.0** | **68.8** | **47.0** | 77.5 | **54.0** | 75.5 | 56.0 | 29.4 | **63.0** | **71.3** | **47.4** |
| Gemini-3-Pro | 78.7 | **39.4** | 67.2 | 40.8 | 80.3 | 34.3 | **81.3** | 42.0 | **76.4** | 37.9 | 59.5 | 40.8 | 75.5 | 41.0 | **78.6** | **62.6** | **35.4** | 49.5 | 70.3 | 43.2 |
| *Open-source LLMs* | | | | | | | | | | | | | | | | | | | | |
| WebSynthesis-8B | 24.2 | 13.0 | 6.9 | 8.0 | 22.5 | 4.0 | 73.2 | 33.0 | 6.8 | 1.0 | 0.4 | 0.0 | 10.5 | 9.0 | 2.6 | 4.0 | 3.4 | 7.0 | 16.7 | 8.8 |
| WMA-8B | 15.2 | 11.0 | 9.4 | 6.0 | 18.8 | 5.0 | 7.5 | 4.0 | 5.4 | 2.0 | 1.2 | 1.0 | 9.8 | 7.0 | 4.5 | 3.0 | 28.5 | 39.0 | 11.1 | 8.7 |
| Word2World-8B | 8.5 | 1.0 | 6.5 | 0.5 | 13.5 | 0.8 | 4.5 | 1.0 | 3.0 | 0.0 | 2.5 | 0.0 | 4.0 | 0.0 | 3.5 | 0.0 | 17.0 | 2.0 | 7.0 | 0.6 |
| Qwen3-8B | 41.4 | 18.0 | 18.5 | 15.0 | 60.4 | 19.0 | 34.0 | 11.0 | 16.8 | 11.0 | 11.5 | 5.0 | 19.8 | 12.0 | 11.2 | 20.0 | 28.8 | 46.0 | 26.9 | 17.4 |
| Qwen3-14B | 49.4 | 25.0 | 31.7 | 23.0 | 71.2 | 25.0 | 51.7 | 13.0 | 30.3 | 14.0 | 25.9 | 12.0 | 31.2 | 18.0 | 17.4 | 24.0 | 33.0 | 50.0 | 38.0 | 22.7 |
| Qwen3-32B | 52.9 | 21.0 | 34.9 | 26.0 | 71.2 | 23.0 | 47.7 | 19.0 | 32.5 | 15.0 | 22.3 | 11.0 | 46.3 | 21.0 | 23.0 | 23.0 | 29.9 | 48.0 | 40.1 | 23.0 |
| *Ours* | | | | | | | | | | | | | | | | | | | | |
| **WebWorld-8B** | 76.7 | 34.0 | 68.0 | 42.0 | 81.7 | **45.0** | **82.2** | 43.0 | 70.3 | 41.0 | **65.9** | 39.0 | 75.5 | 44.0 | 72.7 | **51.0** | 37.6 | 41.0 | 70.1 | 42.2 |
| **WebWorld-14B** | 76.1 | 36.0 | 74.0 | 50.0 | **87.7** | 41.0 | 79.8 | 44.0 | **73.3** | 45.0 | 62.7 | **41.0** | 71.4 | 47.0 | 73.2 | 49.0 | 38.1 | 49.0 | 70.7 | 44.7 |
| **WebWorld-32B** | **77.0** | **37.0** | **74.5** | **51.0** | 87.0 | 40.0 | 79.0 | **44.5** | 73.0 | **45.5** | 63.0 | 40.5 | 73.0 | **48.0** | 74.0 | 50.0 | **38.5** | **54.0** | 71.0 | 45.6 |

**Fact.**: Factuality Score (measures functional correctness of state transitions). **Tur.**: Web Turing Score (measures perceptual realism via adversarial discrimination). Scores are presented as percentages (0–100) for improved readability.

*Table 4.* **Judge Consistency.** We evaluate the consistency of model rankings using two state-of-the-art judges: **GPT-4o** and **Claude-Opus-4.1**. Despite variations in absolute strictness, the relative ranking remains robust across different evaluators.

| Model | GPT-4o | | Claude-Opus-4.1 | | Avg. | |
|---|---|---|---|---|---|---|
| | Fact. | Turing | Fact. | Turing | Fact. | Turing |
| *Proprietary* | | | | | | |
| GPT-4o | 59.5 | 35.4 | 51.6 | 21.0 | 55.6 | 28.2 |
| Claude-Sonnet-4.5 | 59.7 | 36.2 | 59.9 | 31.9 | 59.8 | 34.1 |
| Gemini-3-Pro | 70.3 | 43.2 | 72.8 | 36.5 | 71.6 | 39.9 |
| *Open-weights* | | | | | | |
| Qwen3-8B | 26.9 | 17.4 | 27.3 | 11.7 | 27.1 | 14.6 |
| Qwen3-14B | 38.0 | 22.7 | 35.3 | 15.8 | 36.7 | 19.3 |
| Qwen3-32B | 40.1 | 23.0 | 37.0 | 16.2 | 38.6 | 19.6 |
| *Ours* | | | | | | |
| **WebWorld-8B** | **70.1** | **42.2** | **67.6** | **31.7** | **68.9** | **37.0** |

also validate practical utility through extrinsic evaluation, measuring downstream agent performance when trained on WebWorld-synthesized data (Section 5.1).

## 4.1. Metrics

To provide a holistic assessment of the world model's capabilities, we evaluate performance across nine dimensions using two metrics. Both metrics utilize GPT-4o as a judge to automate the evaluation process.

**Factuality Score.** This metric evaluates the functional correctness of the state transitions via pointwise scoring. Given the interaction history and the ground-truth next state, the judge assesses whether the model's predicted observation accurately reflects the causal effect of the action (e.g., a button click triggering a pop-up). The complete judge prompt is provided in Appendix 13. The score quantifies how well the model avoids hallucinations and aligns with the deterministic dynamics of the real web, focusing on semantic consistency rather than pixel-perfect matching.

**Web Turing Score.** This metric evaluates the world model through pairwise comparison. We present the judge with two anonymized observations—one generated by Web-World and one from the real browser environment—and ask it to identify the more realistic webpage (see Appendix 14 for the full prompt). A higher score indicates that the model's generated states are indistinguishable from, or even deemed more plausible than, real-world data.

## 4.2. Evaluation Dimensions

We constructed WebWorld-Bench, a comprehensive evaluation suite comprising nine distinct dimensions. The evaluation data were generated using the same hierarchical data curation pipeline as our training set to ensure domain alignment, but were strictly held out to prevent data contamination. **Long-Horizon Consistency** evaluates context retention in extended interactions. We select trajectories exceeding 10 steps and provide the full interaction history as input. **Fine-Grained Sensitivity** challenges the model's precision by focusing on localized state updates. We employ an LLM to specifically filter for actions that trigger minimal changes—such as expanding a dropdown menu or toggling a checkbox—requiring the model to accurately localize up-

*Table 5.* **Downstream Performance.** Comparison of the base **Qwen3-8B and Qwen3-14B models** versus **the models** fine-tuned on WebWorld-synthesized trajectories. We report **Success Rate (SR %)**, Standard Error (Std), and Average Steps. **The symbols ↑ and ↓ indicate that higher and lower values are better, respectively.** For WebArena, we detail performance across sub-domains.

| Model | MiniWob++ | | | WebArena | | | | | | |
|---|---|---|---|---|---|---|---|---|---|---|
| | SR ↑ | Std | Steps ↓ | SR ↑ | Std | Steps ↓ | Domain Breakdown (SR %) | | | |
| | (%) | (±) | (Avg) | (%) | (±) | (Avg) | E-Comm | GitLab | Reddit | Others |
| GPT-4o | **64.3** | 0.019 | 4.12 | 26.6 | 0.016 | 11.96 | 26.8 | 27.5 | 24.5 | 25.3 |
| Qwen3-8B (Base) | 49.4 | 0.020 | 4.88 | 9.8 | 0.018 | 15.24 | 17.1 | 9.4 | 5.0 | 18.2 |
| **Qwen3-8B + WebWorld (Ours)** | 59.3 | 0.020 | 4.39 | **20.7** | 0.014 | 19.25 | **20.7** | **21.4** | **23.3** | 17.5 |
| *Improvement (8B)* | *+9.9%* | *-* | *-0.49* | *+10.9%* | *-* | *+4.01* | *+3.6%* | *+12.0%* | *+18.3%* | *-0.7%* |
| Qwen3-14B (Base) | 54.9 | 0.020 | 4.55 | 15.1 | 0.013 | 16.12 | 15.4 | 15.4 | 6.3 | 18.3 |
| **Qwen3-14B + WebWorld (Ours)** | 63.2 | 0.019 | 4.28 | **24.3** | 0.015 | 17.11 | **24.0** | **32.7** | **21.0** | 17.6 |
| *Improvement (14B)* | *+8.3%* | *-* | *-0.27* | *+9.2%* | *-* | *+0.99* | *+8.6%* | *+17.3%* | *+14.7%* | *-0.7%* |

dates without hallucinating global shifts. Conversely, **Base Semantics** assesses performance on macroscopic page transitions. Finally, to ensure the model learns generalized dynamics rather than specific syntax, we evaluate **Format Robustness** across multiple representations (HTML, XML, Markdown) and **Web2NAL**, which tests the model's ability to verbally describe state changes in natural language.

### 4.3. Result on WebWorld-Bench

Table 3 shows that WebWorld-32B achieves 71.0% average Factuality Score, matching Claude-Opus-4.1 (71.3%), with particularly strong long-horizon consistency (77.0%) and multi-format robustness (70–75% across formats). The notably low scores of open-source baselines reflect output format misalignment rather than model deficiency; detailed analysis is provided in Appendix C.

### 4.4. Judge Consistency

To ensure that the performance ranking in WebWorld is robust across different judge models, we conduct a consistency analysis using different LLMs as judges. We measure the Total Score across the test set for each judge. As shown in Table 4, while absolute scores may vary, the relative ranking of models remains consistent.

## 5. Extrinsic Evaluation

### 5.1. Trajectory Synthesis for Agents

We evaluate whether synthetic data from WebWorld improves real-world agent benchmarks. We implement an **Abstract-and-Instantiate** data synthesis pipeline to scale up training examples, generating 8,000 trajectories. The pipeline works as follows: Starting with concrete seed tasks (e.g., "Book a flight to London on March 15th"), we use an LLM to **abstract** them into underspecified goals (e.g.,

*Table 6.* **Inference-Time Lookahead Search on MiniWob. Fmt**: NL = Natural Language, A11y = A11y Tree. **Scoring**: Point = Pointwise, Pair = Pairwise. **Alg**: BoN = Best-of-$N$.

| Model & Search Configuration | | | | | Result | |
|---|---|---|---|---|---|---|
| World Model | Value Model | Fmt | Score | Alg ($k$) | Reward | Δ |
| *Baselines* | | | | | | |
| - | - | - | - | Greedy | 64.3 | - |
| GPT-4o | GPT-4o | A11y | Point | BoN (3) | 63.8 | -0.5% |
| *Impact of Scoring & Value Model* | | | | | | |
| **Ours (WebWorld)** | GPT-4o | A11y | Point | BoN (3) | 64.8 | +0.5% |
| GPT-5 | GPT-4o | A11y | **Pair** | BoN (3) | 64.5 | +0.2% |
| **Ours (WebWorld)** | GPT-4o | A11y | **Pair** | BoN (3) | **65.5** | +1.2% |
| Ours (WebWorld) | **GPT-5** | A11y | Pair | BoN (3) | **67.5** | +3.2% |
| *Format & Context Trade-off* | | | | | | |
| Ours (WebWorld) | GPT-4o | NL | Pair | BoN (3) | 65.2 | +0.9% |
| Ours (WebWorld) | GPT-4o | **NL** | Pair | BoN (5) | **65.9** | +1.6% |
| Ours (WebWorld) | GPT-4o | **A11y** | Pair | BoN (2) | **65.7** | +1.4% |
| *Advanced Search Strategy* | | | | | | |
| Ours (WebWorld) | GPT-4o | A11y | Pair | **MCTS (3)** | 65.4 | +1.1% |
| Ours (WebWorld) | GPT-4o | A11y | Pair | **Hybrid (3)** | 65.5 | +1.2% |

"Book a flight to somewhere on sometime"). The agent then executes actions in WebWorld following these abstract goals. For each execution trajectory, we **instantiate** it back into a concrete task. Finally, the agent conducts the concrete task, and we apply rejection sampling to retain only successful trajectories. We fine-tuned Qwen3-8B on this synthetic dataset. As shown in Table 5, the model achieves significant improvements over the base model, with a **9.9%** gain on MiniWob++ and a **10.9%** gain on WebArena. Reddit and GitLab show strong gains of **18.3%** and **12.0%**.

### 5.2. Inference-Time Search with World Models

We implement the lookahead search to validate WebWorld's utility, following Gu et al. (2025) and Chae et al. (2025). At each step, the agent proposes $N$ candidate actions. For each candidate, WebWorld simulates the next state. A value

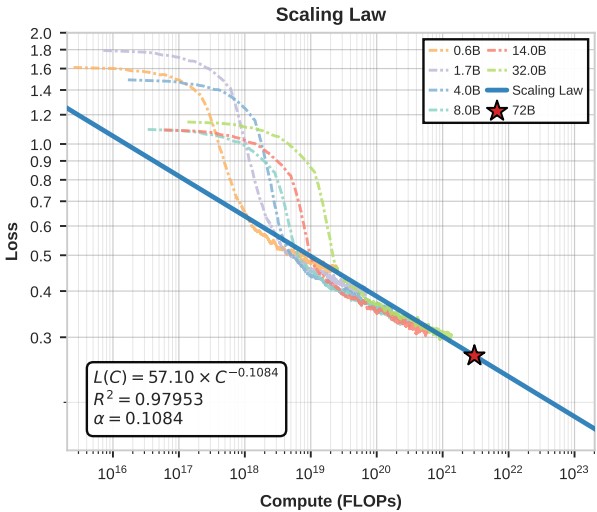

*Figure 4.* **Scaling Law of WebWorld.** Larger models achieve lower eval loss. Stars indicate predictions for the 72B model, suggesting continued performance gains with model scaling.

model then evaluates these simulated states for task utility, and the agent executes the action with the highest score. As detailed in Table 6, our WebWorld as a world model achieves better performance than GPT-5. For the value model, shifting from pointwise to pairwise evaluation yields substantial gains. For the output format, natural language outputs enable deeper planning ($k = 5$) while full HTML is restricted to shallow depths ($k = 2$) by context limits. For search strategy, advanced MCTS or hybrid search (which only triggers look-ahead when actions are uncertain) offer marginal improvement. Consequently, the bounded gains from inference-time search suggest that **world models are more valuable for synthesizing training data**, aligning with Qian et al. (2026)'s observation that current agents derive limited benefit from inference-time search.

## 6. Analysis

### 6.1. Scaling Law of WebWorld

We trained the WebWorld across 6 model sizes using identical settings. Figure 4 shows that larger models consistently achieve lower evaluation loss. The relationship between the final evaluation loss $L$ and compute $C$ (measured in FLOPs) follows a power-law. We extrapolate predictions for 72B models (marked with stars in Figure 4). The predicted losses suggest substantial performance improvements are achievable through further scaling, **with no signs of saturation.**

### 6.2. Ablation of Reasoning Activation

We test the impact of different amounts of CoT data on model performance. As shown in Table 7, a minimal

*Table 7.* **Reasoning Activation Ablation.** Comparison under varying reasoning data scales. **WebWorld-8B achieves superior performance with only 1k samples**.

| Model Source | Data Scale | Factuality Score | Web Turing Score | Total Score |
|---|---|---|---|---|
| *Teacher Model* | | | | |
| Claude-Opus-4.1 | - | 0.713 | 0.474 | 0.594 |
| *From Qwen3-8B (Direct Reasoning Tuning)* | | | | |
| | 500 | 0.502 | 0.277 | 0.390 |
| Qwen3-8B | 1k | 0.511 | 0.296 | 0.403 |
| | 2k | 0.541 | 0.319 | 0.430 |
| | 10k | 0.625 | 0.394 | 0.510 |
| *From Stage 1 Model (1.06M Transition Modeling)* | | | | |
| Ours | 500 | 0.668 | 0.402 | 0.535 |
| **Ours** | **1k** | **0.701** | **0.422** | **0.561** |
| Ours | 2k | 0.686 | 0.388 | 0.537 |
| Ours | 10k | 0.692 | 0.413 | 0.552 |

*Table 8.* **Cross-Environment Adaptation.** WebWorld demonstrates strong adaptability across unseen environments.

| Environment | 1,500 Samples *(Total Score)* | | | 3,000 Samples *(Total Score)* | | |
|---|---|---|---|---|---|---|
| | Qwen3 | **Ours** | **Gain ($\Delta$)** | Qwen3 | **Ours** | **Gain ($\Delta$)** |
| API Services | 0.088 | **0.299** | +0.211 | 0.258 | **0.292** | +0.034 |
| Code | 0.147 | **0.396** | +0.249 | 0.196 | **0.471** | +0.275 |
| Game | 0.253 | **0.473** | +0.220 | 0.374 | **0.522** | +0.148 |
| GUI | 0.322 | **0.705** | +0.383 | 0.511 | **0.719** | +0.208 |
| **Average** | 0.176 | **0.400** | **+0.224** | 0.298 | **0.463** | **+0.165** |

dataset of just 1,000 samples is sufficient to activate the reasoning pattern, yielding a Total Score of 0.561—surpassing the base model trained on 10x more data (0.510) and approaching the teacher model (0.594). We observe that excessive CoT data can degrade performance, thus **we recommend combining large-scale real-world training with a small, carefully curated amount of CoT data for optimal results.**

### 6.3. Cross-Environment Generalization

We evaluate WebWorld's adaptation capability by fine-tuning on open-source agent trajectories from API services, code development, games, and GUI desktops, converted into $(s_t, a_t, s_{t+1})$ transition tuples (Tables 14 and 15), using the same Factuality and Web Turing metrics from WebWorld-Bench. Results in Table 8 show that WebWorld consistently outperforms the baseline, demonstrating strong transferability across other environments.

## 7. Conclusions and Limitations

In this paper, we introduced WebWorld, a browser simulator trained on over one million real-world interaction trajectories. WebWorld enables efficient agent training in simulation, significantly improving performance on downstream

tasks. WebWorld has limitations. It exhibits sycophancy by generating overly optimistic outcomes that cater to the agent's action. Additionally, WebWorld struggles to generate high-quality, detailed content, such as scientific articles.

## Impact Statement

This paper presents WebWorld, a large-scale world model designed to simulate web environments for training autonomous agents. Our work aims to advance the field of machine learning by enabling scalable, offline training that circumvents the latency, safety constraints, and rate-limiting issues inherent to real-world web interaction.

The primary societal benefit of this work is the democratization of web agent research. By providing an open, high-fidelity simulator trained on diverse real-world trajectories, we lower the barrier to entry for developing capable web agents, which can improve digital productivity, automate repetitive tasks, and enhance web accessibility for users with disabilities. Moreover, training in simulation mitigates safety risks: agents can explore without triggering irreversible real-world consequences such as unintended purchases, form submissions, or data modifications.

We acknowledge that more capable web agents introduce dual-use concerns. Malicious actors could leverage such agents for automated phishing campaigns, credential stuffing, or large-scale scraping that violates terms of service. Additionally, our training data is sourced from web crawls (FineWeb, CCI 3.0), which—despite rigorous keyword and LLM-based filtering—may inadvertently contain personally identifiable information (PII), toxic content, or demographic biases reflected in public web data. While we strictly adhere to `robots.txt` protocols and apply safety heuristics, residual risks remain. We also observe a sycophancy bias in the model's predictions, where simulated outcomes can be overly optimistic or cater to agent expectations, potentially hindering robust policy learning.

To address these concerns, we release WebWorld with comprehensive documentation and ethical guidelines. We encourage the community to build on this work by developing PII scrubbing techniques, adversarial robustness mechanisms, and alignment methods to reduce sycophancy. We recommend that practitioners apply additional domain-specific safety checks before deploying agents trained on WebWorld in high-stakes environments. By open-sourcing both the model and training pipeline, we aim to foster transparent, reproducible research that prioritizes safety alongside capability advancement.

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

# Appendix Contents

# A. World Model Training Details

We utilize LLaMA-Factory (Zheng et al., 2024) for the supervised fine-tuning (SFT) of our LLM-based world model. We employ full fine-tuning with DeepSpeed ZeRO-3 (Qwen3-32B) and ZeRO-2 (others), enabling Liger Kernel and Unsloth garbage collection for memory efficiency. We apply sequence packing with a maximum cutoff length of 20,000 tokens to optimize data throughput. The training process runs for 1 epoch with a cosine learning rate scheduler. The detailed hyperparameters are in Table 9, which provides a side-by-side comparison of the hyperparameters used in our two-stage training curriculum.

*Table 9.* **Comparison of Hyperparameters Across Training Stages.** Stage 1 focuses on large-scale dynamics learning with aggressive data throughput, while Stage 2 refines reasoning capabilities with conservative fine-tuning to prevent forgetting. Training times are reported for three model scales on NVIDIA A100 GPUs.

| Hyperparameter | Stage 1: Transition Modeling | Stage 2: Reasoning Activation |
|---|---|---|
| ***Training Configuration*** | | |
| Objective | Learn $(s_t, a_t) \rightarrow s_{t+1}$ | Learn $(s_t, a_t) \rightarrow$ thought $\rightarrow s_{t+1}$ |
| Data Source | 1.06M real-world trajectories | 1K CoT samples |
| Initialization | Qwen3 Base Checkpoint | Stage 1 Checkpoint |
| Finetuning Type | Full Parameter | Full Parameter |
| Precision | BF16 | BF16 |
| Optimization | DeepSpeed ZeRO-2 | DeepSpeed ZeRO-2 |
| ***Learning Rate Schedule*** | | |
| Base Learning Rate | $2.0 \times 10^{-5}$ | $8.0 \times 10^{-6}$ (↓ 2.5×) |
| Scheduler Type | Cosine with Warmup | Cosine with Warmup |
| Warmup Ratio | 0.1 | 0.1 |
| Number of Epochs | 1 | 5 (↑ 5×) |
| ***Batch Configuration*** | | |
| Per-Device Batch Size | 2 | 2 |
| Gradient Accumulation | 2 steps | 2 steps |
| Effective Batch Size | 64 (2 devices) | 32 (1 device) |
| ***Data Processing*** | | |
| Max Sequence Length | 20,000 tokens | 20,000 tokens |
| Sequence Packing | Enabled | Enabled |
| History Masking | Disabled | **Enabled** (CoT-only loss) |
| ***Training Resources (WebWorld-8B)*** | | |
| Hardware Configuration | 16×A100 (80GB) | 8×A100 (80GB) |
| Total Training Steps | ∼7,215 steps | ∼110 steps |
| ***Training Time by Model Scale*** | | |
| WebWorld-8B | 4 days, 1:12:13 (16×A100) | 2:02:54 (8×A100) |
| WebWorld-14B | 7 days, 21:50:56 (16×A100) | 2:25:03 (8×A100) |
| WebWorld-32B | 12 days, 20:20:06 (16×A100) | 3:07:03 (8×A100) |

We utilize the Qwen3 series (Yang et al., 2025) as the primary backbone for our LLM-based world models. To systematically study the impact of model scale on world modeling capabilities, we train Qwen3 models across three varying parameter sizes: 8B, 14B, and 32B.

# B. API Models

We evaluate our methods using state-of-the-art proprietary models. The specific API models and their corresponding versions used in our experiments are listed in Table 10.

# C. Baseline Implementation Details

**WMA Baseline Construction.** For the WMA baseline (Chae et al., 2025), the official release provides only a LoRA adapter compatible with `Meta-Llama-3.1-8B-Instruct`, which precludes a direct architectural comparison with our

*Table 10.* API models and versions used for evaluations.

| Model | Version |
|---|---|
| GPT-4o | `gpt-4o-2024-11-20` |
| Claude-Sonnet-4.5 | `claude-sonnet-4-5-20250929` |
| Claude-Opus-4.1 | `claude-opus-4-1-20250805` |
| Gemini-3-Pro | `Gemini-3-Pro-preview` |
| GPT-5 | `gpt-5-2025-08-07` |

Qwen3-based models. To ensure a fair evaluation controlled for the base model, we utilized the official WMA dataset[1]. We reformatted this data to align with our unified training schema and performed fine-tuning on `Qwen3-8B` using the converted dataset. This approach allows us to evaluate the efficacy of the data and training objective independently of the underlying foundation model. **WMA's core objective is predicting free-form natural language descriptions of state changes rather than structured state representations.**

**WebSynthesis Baseline Construction.** For the WebSynthesis baseline (Gao et al., 2025), as the pre-trained world model weights were not publicly released, we reproduced the model using their official open-source dataset[2]. Specifically, we utilized the `world-model-training-data-27k.json` subset. We reformatted these samples to match our unified training template and fine-tuned `Qwen3-8B` under the exact same hyperparameters as our main experiments. This ensures that the comparison isolates the impact of the training data distribution and quality. **WebSynthesis is optimized for sparse, multi-page transitions characteristic of WebArena (e.g., post-form-submission page loads).**

**Word2World Baseline Construction.** For the Word2World baseline (Li et al., 2025b), we utilized the open-weights checkpoint `WorldModel-Webshop-Llama3.1-8B`[3]. Word2World adopts a simplified, flattened text stream representation with custom separators, which is structurally distinct from the standard A11y Tree or HTML formats. To benchmark its performance, we evaluated the model in a zero-shot setting on WebWorld-Bench. As expected, the significant format misalignment—specifically the lack of standard A11y or natural language outputs—resulted in near-zero performance across structural and semantic metrics, highlighting the necessity of format-aligned training for robust web simulation. **Word2World's proprietary WebShop format is not transferable to open-domain evaluation.**

## D. Training Data Composition

We provide comprehensive statistics of all datasets used in Stage 1 training (Real-World Transition Modeling) in Table 11. The table details each dataset's category, original source, language distribution (English/Chinese/Multilingual), scale, and key attributes.

## E. Domain Distribution Statistics

The chart illustrates the distribution of domains and data sources in our dataset of over one million trajectories, as shown in Figure 5. The colors represent different semantic domains (e.g., Technology, E-Commerce), showing that our data collection pipelines significantly contribute to the diversity of open-domain topics compared to traditional web generation methods.

## F. Action Space Definition

To enable the agent to interact effectively across diverse web environments—ranging from DOM-based websites to coordinate-sensitive mini-games—we define a unified action space represented as Python-style function calls. The action space is hybrid, supporting both high-level semantic interactions via element identifiers (A11y Tree IDs, denoted as `bid`) and low-level control via Cartesian coordinates ($(x, y)$). Element-based actions allow precise manipulation of form fields, buttons, and dropdowns, while coordinate-based primitives (e.g., `mouse_move`, `mouse_click`) enable the agent to handle

---

[1] https://huggingface.co/datasets/LangAGI-Lab/world_model_for_wa_desc_with_tao_formatted_w_cot
[2] https://huggingface.co/datasets/yifeigao/WebSynthesis
[3] https://huggingface.co/X1AOX1A/WorldModel-Webshop-Llama3.1-8B

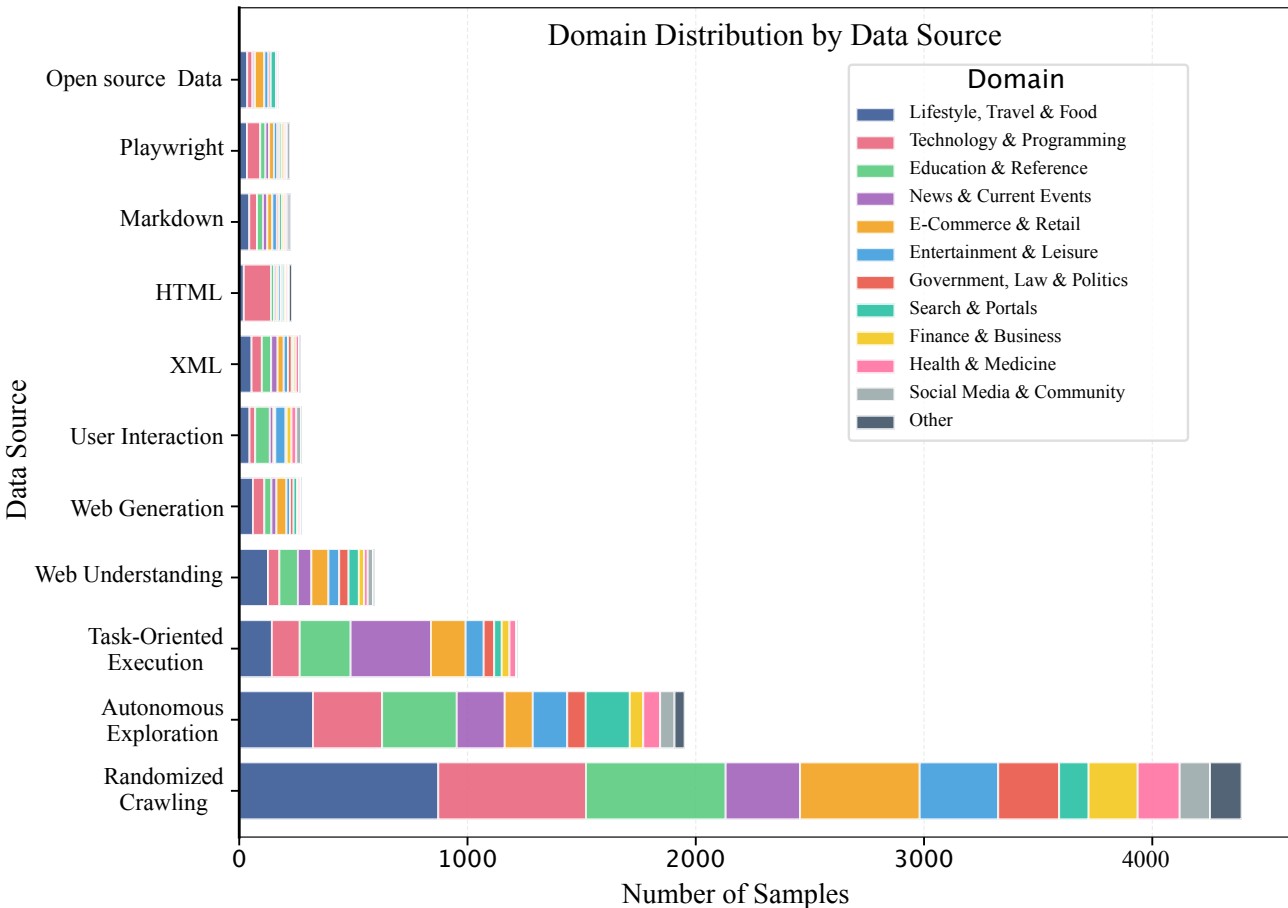

*Figure 5.* **Domain and Source Distribution of WebWorld Training Data.** The chart illustrates the composition of our trajectory dataset, which contains over one million samples, across 15 distinct data sources. The colors represent different semantic domains (e.g., Technology, E-Commerce), showing that our data collection pipelines significantly contribute to the diversity of open-domain topics compared to traditional web generation methods.

*Table 11.* **Statistics of the Constructed WebWorld Training Data.** Attributes indicate if data is from Real-world websites, contains Long-horizon sequences (>10 steps), or supports Multi-format (HTML/A11y).

| Category | Source | Lang. | Scale | | Attributes | | |
|---|---|---|---|---|---|---|---|
| | | | # Traj. | Size | Real-World | Long-Seq. | Multi-Fmt. |
| **Randomized Crawling** | CCI 3.0 (Filtered) | CN | 57,837 | 17.67G | ✓ | × | × |
| | FineWeb Subset | EN | 235,674 | 4.33G | ✓ | × | × |
| | *Subtotal* | - | **293,511** | *22.0G* | | - | |
| **Autonomous Exploration** | FineWeb (LLM-Driven) | Mixed | 36,474 | 9.9G | ✓ | ✓ | × |
| | High-Freq Sites | Mixed | 1,882 | 0.5G | ✓ | ✓ | × |
| | *Subtotal* | - | **38,356** | *10.4G* | | - | |
| **Task-Oriented Execution** | Benchmarks (Synthetic) | Mixed | **94,001** | **8.2G** | × | ✓ | ✓ |
| **Open Source** | AgentTrek / OS_Gen / Etc. | EN | 37,568 | 0.7G | ✓ | ✓ | × |
| **Multi-format** | HTML / XML / Playwright | - | 47,855 | 4.0G | ✓ | × | ✓ |
| **Interaction** | Ultrachat / QA / Web2NAL | - | 547,758 | 5.6G | × | × | × |
| **Total** | **All Combined** | - | **1,059,348** | **50.9 G** | ✓ | ✓ | ✓ |

canvas elements or drag-and-drop tasks. Additionally, the agent possesses browser-level controls for navigation and tab management, as well as meta-actions to communicate with the user or terminate the trajectory. The complete set of supported action primitives is detailed in Table 12 and Table 13.

*Table 12.* The unified action space available to the agent. Actions are categorized by interaction type. Parameters include element IDs (`bid`), coordinates ($x, y$), textual input (`text`), and navigation deltas.

| Category | Action Primitives | Description |
|---|---|---|
| **Element Interactions** | `click(bid, button, mods)` | Click a specific DOM element identified by `bid`. |
| | `fill(bid, text, auto)` | Input text into a focused field (supports autocomplete). |
| | `select_option(bid, opts)` | Select single or multiple options from a dropdown/combobox. |
| | `hover(bid)` | Hover the cursor over a specific element. |
| **Coordinate & Mouse** | `mouse_move(x, y)` | Move cursor to absolute screen coordinates. |
| | `mouse_click(x, y, button)` | Click at a specific coordinate (supports double click). |
| | `mouse_{down, up}(x, y)` | Hold or release mouse button (enables drag-and-drop). |
| **Keyboard** | `keyboard_press(key)` | Press a specific physical key (e.g., 'Enter', 'Tab'). |
| | `keyboard_type(text)` | Type a string of text sequentially. |
| **Browser & Nav.** | `scroll(dx, dy)` | Scroll the viewport horizontally or vertically. |
| | `goto(url), go_{back, fwd}` | Navigate to a URL or traverse history stack. |
| | `tab_{new, close, focus}` | Manage browser tabs (open, close, or switch focus). |
| **Meta & Control** | `send_msg_to_user(text)` | Output a message to the user (e.g., for clarification). |
| | `noop(wait), infeasible` | Wait for a duration or declare the task impossible. |

## G. Format Conversion Pipeline

To train a robust and generalizable browser world model, we design a unified data format that balances structural fidelity, token efficiency, and cross-domain transferability. Our format choice is driven by three core principles: *universal applicability*, *LLM compatibility*, and *multi-format robustness*.

**A11y Tree as Primary Representation.** We adopt the **A11y Tree** as our primary state representation for browser interactions. Unlike raw HTML, which contains rendering noise (CSS, scripts, layout metadata), the A11y Tree provides a structured,

*Table 13.* Action Distribution by Category

| Category | Action Primitive | Percentage | Category | Action Primitive | Percentage |
|---|---|---|---|---|---|
| *Element Interactions* | click | 77.29% | *Browser & Navigation* | scroll | 0.88% |
| | fill | 5.12% | | goto | 10.06% |
| | select_option | 0.96% | | go_back | 0.24% |
| | hover | 0.06% | | go_fwd | 0.15% |
| | *Subtotal* | *83.43%* | | tab_new | 0.22% |
| *Coordinate & Mouse* | mouse_move | 0.42% | | tab_close | 0.19% |
| | mouse_click | 0.35% | | tab_focus | 0.15% |
| | mouse_down | 0.18% | | *Subtotal* | *11.89%* |
| | mouse_up | 0.18% | *Meta & Control* | send_msg_to_user | 1.34% |
| | *Subtotal* | *1.13%* | | noop | 0.74% |
| *Keyboard* | keyboard_press | 0.55% | | infeasible | 0.30% |
| | keyboard_type | 0.62% | | *Subtotal* | *2.38%* |
| | *Subtotal* | *1.17%* | | | |
| **Total (20 actions): 100.00%** | | | | | |

hierarchical abstraction of interactable UI elements with high information density. This format has proven effective across diverse digital environments—not only for web tasks but also for GUI automation on Android, macOS, and Linux systems (de Chezelles et al., 2025). Empirically, A11y Tree achieves decent token compression compared to raw HTML while preserving all action-critical semantics, making it the de facto standard for text-based web agent research (Zhou et al., 2023; Deng et al., 2023). We extract A11y Trees using the Playwright API from the BrowserGym framework (de Chezelles et al., 2025), which exposes browser accessibility layers in a consistent format across Chromium, Firefox, and WebKit engines. Each node in the tree encodes its role (e.g., `button`, `textbox`), properties (e.g., `focused`, `required`), and a unique identifier (`bid`) for action grounding.

**Representation Transformation Process** The conversion pipeline employs a two-stage parse-then-generate architecture to transform web A11y Tree into multiple target representations. In the parsing stage, the `AccessibilityTreeParser` converts the indentation-based textual format into a canonical nested dictionary structure by applying regular expressions to extract node attributes (role, name, ID, properties) and utilizing a stack-based algorithm to reconstruct the hierarchical parent-child relationships from indentation levels. This intermediate representation serves as the single source of truth for all subsequent transformations. In the generation stage, format-specific generators (`XMLGenerator`, `HTMLGenerator`, `PlaywrightGenerator`, `MarkdownGenerator`) traverse the parsed tree structure and apply domain-specific mapping rules—such as ARIA-to-HTML semantic mapping, XML name sanitization, or Playwright reference ID assignment—to produce target-format outputs. The entire process operates on JSONL conversation files by identifying A11y Tree segments delimited by markers (`"Initial Page State:"` and `"First Action:"`), transforming only the extracted tree content while preserving surrounding conversational context, thereby enabling efficient batch preprocessing of web interaction datasets for downstream tasks such as UI automation testing, agent training, and accessibility analysis.

## H. URL Filtering with LLMs

To ensure high-quality and safe data sources, we apply LLM-based filtering to candidate URLs extracted from pre-training corpora. As shown in Figure 6, we evaluate each URL across four dimensions: *accessibility* (page reachability), *content suitability* (absence of unsafe content), *interactivity* (presence of actionable elements), and *engineering quality* (HTML structural soundness). An LLM judge assigns scores (0–1) for each dimension. Low-scoring URLs are filtered out, retaining 85.2% of the candidates that passed initial rule-based checks.

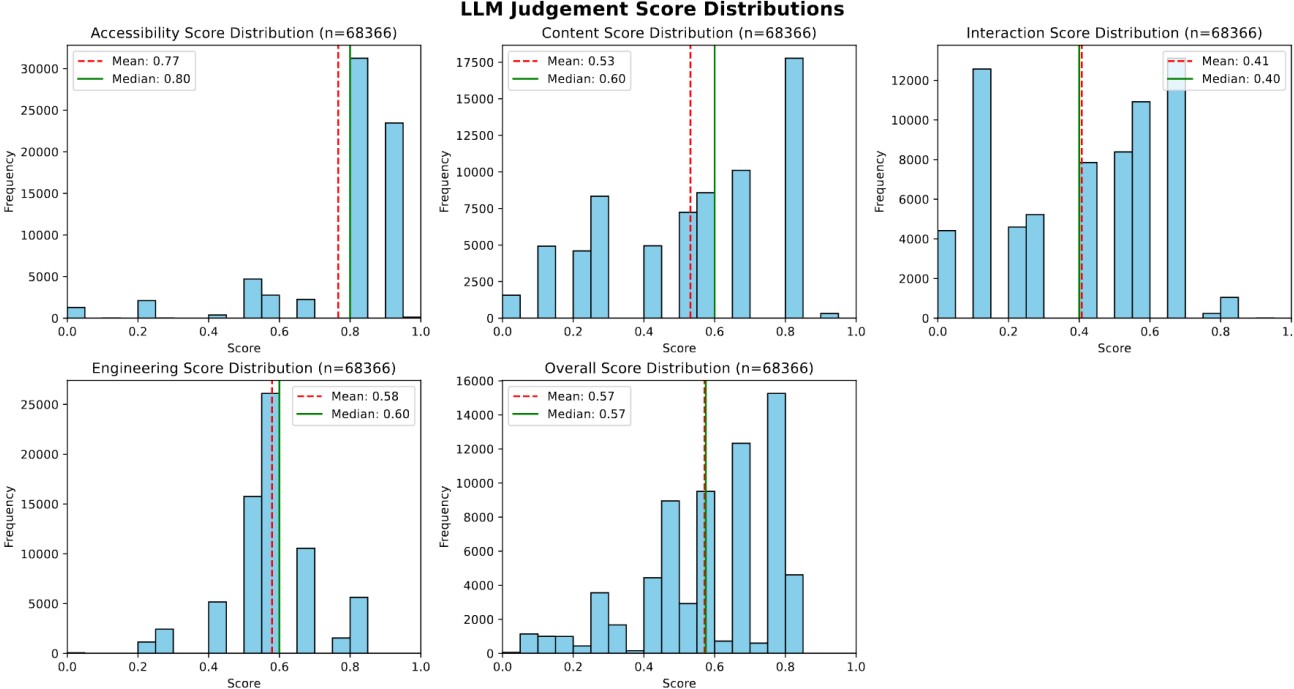

*Figure 6.* **LLM-based URL Filtering Pipeline.** We evaluate candidate URLs across four dimensions—accessibility, content suitability, interactivity, and engineering quality—using an LLM judge. The chart shows the distribution of scores and the filtering threshold (red dashed line), which retains only the top 32% of URLs for data collection.

## I. Cross-Environment Generalization Data

Since no standardized world model benchmarks exist for non-web environments, we construct training and test sets for four domains (API services, code, games, GUI) by collecting open-source agent trajectories and converting them into $(s_t, a_t, s_{t+1})$ transition tuples following WebWorld's data format. Dataset statistics are provided in Tables 14 and 15.

## J. World Model Evaluation Taxonomy

Existing work on world models for web and GUI agents can be categorized based on their evaluation strategies: *intrinsic evaluation* approaches that explicitly measure world model quality, and *extrinsic evaluation* approaches that assess world models through downstream task performance.

**Intrinsic Evaluation.** WebEvolver (Fang et al., 2025) evaluates world models along three dimensions: structural correctness, content similarity, and functional/semantic consistency between predicted and actual web states. Similarly, WMA (Chae et al., 2025) adopts an information coverage metric, measuring the overlap between predicted state change descriptions and ground truth using ROUGE and BERTScore. These approaches collect training data in the WebArena environment, with WebEvolver using MCP Playwright to gather A11y Tree pairs $(A11y_t, A11y_{t+1})$, while WMA employs GPT-4o to collect high-quality trajectories and uses the Hungarian algorithm for DOM diffing, subsequently converting diffs to natural language descriptions via LLM. ViMo (Anonymous, 2025b), focusing on mobile app GUIs, proposes a more comprehensive evaluation framework with four metrics: visual similarity (sgc), instruction accuracy (sia), functional availability (sar), and user studies. The work leverages existing large-scale GUI interaction datasets such as AITW.

**Extrinsic Evaluation.** WebDreamer (Gu et al., 2025) and WebSynthesis (Gao et al., 2025) evaluate world models extrinsically through end-to-end task success rates. WebDreamer randomly explores real-world websites and uses vision-language models (VLMs) to synthesize descriptive labels for $(screenshot_{before}, action, screenshot_{after})$ triplets. WebSynthesis collects $(A11y_{t-1}, Action_t, A11y_t)$ triplets in WebArena through random exploration, integrating the world model into Monte Carlo Tree Search (MCTS) where model quality is reflected in final agent performance. While extrinsic evaluation provides practical insights into world model utility, it conflates world model quality with other agent components, making it

*Table 14.* Overview of the Training Set in Cross-Environment Data (training set).

| Category | Rank | Dataset Name | Project Source | Environment Type | Samples |
|---|---|---|---|---|---|
| **A. Code & Development** | 1 | `wm_intercode_sql.jsonl` | AgentBank/intercode_sql | code/IDE, API | 4,522 |
| | 5 | `wm_full_sft.jsonl` | Neulab/swe-smith | terminal/shell, code/IDE | 17,380 |
| | 8 | `wm_train-00005-of-00012.jsonl` | SWE-agent-trajectories | terminal/shell, code/IDE | 6,665 |
| | 21 | `wm_train-00002-of-00012.jsonl` | SWE-agent-trajectories | terminal/shell, code/IDE | 6,661 |
| | 24 | `wm_full_sft.jsonl` | Neulab/agenttuning_db | code/IDE, API | 527 |
| | 13 | `wm_full_sft.jsonl` | Neulab/agenttuning_os | terminal/shell | 195 |
| **B. GUI & Desktop** | 7 | `wm_agentnet_win_mac_18k.jsonl` | AgentNet | GUI/desktop | 17,625 |
| **C. Game & Simulation** | 9 | `wm_alfworld_sft.jsonl` | Agent-ETO | game | 3,119 |
| | 10 | `wm_sciworld_sft.jsonl` | Agent-ETO | game, simulation | 1,483 |
| | 26 | `wm_alfworld.jsonl` | AgentBank/alfworld | game, simulation | 3,321 |
| | 11 | `wm_sciworld_sft.jsonl` | Agent-ETO | game, interactive sim | 1,483 |
| **D. API & Services** | 6 | `wm_train-00001-of-00003.jsonl` | Agent-Ark/Toucan | API, terminal/shell | 4,281 |
| | 15 | `wm_toolbench_react_10p.jsonl` | Agent-FLAN | API, task-based | 2,288 |
| | 12 | `wm_full_sft.jsonl` | Neulab/agenttuning_kg | API, KB query | 305 |
| **Total** | *14 Datasets* | | | | **69,855** |

*Table 15.* Overview of the Training Set in Cross-Environment Data (test set).

| Category | Rank | Dataset Name | Project Source | Environment Type | Samples |
|---|---|---|---|---|---|
| **A. Code & Development** | 27 | `wm_train-00001-of-00001.2.jsonl` | SWE-agent-trajectories | terminal/shell, code | 6,660 |
| | 25 | `wm_mbpp_before.jsonl` | AgentBank/mbpp_before | code/IDE | 707 |
| **B. GUI & Desktop** | 2 | `wm_agentnet_ubuntu_5k.jsonl` | AgentNet | GUI, application | 5,000 |
| **C. Game & Simulation** | 16 | `wm_rearrange.jsonl` | AgentBank/rearrange | game, GUI/desktop | 299 |
| | 29 | `wm_alfworld_sft.jsonl` | Agent-ETO | game | 3,119 |
| **D. API & Services** | 19 | `wm_db.jsonl` | AgentInstruct | API, database | 538 |
| *Unused* | 4 | `wm_full_sft.jsonl` | Neulab/openhands | code/IDE, game... | 121 |
| | 28 | `wm_train-00006-of-00001.2.jsonl` | SWE-agent-trajectories | terminal/shell, code | 6,664 |
| **Total** | *6 Datasets (+2 Unused)* | | | | **16,323** |

difficult to isolate the specific contributions of world modeling.

## K. Generation Length Analysis

To better understand how our two-stage training strategy affects model behavior, we analyze the distribution of output token lengths across different training phases. **Figure 7** shows the comparison between the *Real-World Transition Modeling* baseline (Stage 1) and the *Reasoning Activation* phase (Stage 2) under varying data scales.

We observe a substantial shift in generation patterns between the two stages. The Real-World Transition Modeling stage (grey dashed line) produces considerably longer outputs, as the model attempts to capture comprehensive web state details from raw interaction data. Notably, after applying Reasoning Activation, the average output length decreases by approximately 49.4% despite the introduction of reasoning. This suggests that Stage 2 training does more than add reasoning tokens—it fundamentally restructures the model's prediction pattern. By training on high-quality distilled data, the world model shifts from verbose state reconstruction to concise state prediction, effectively filtering redundant information while preserving essential semantic changes. The curve stabilizes after approximately 1,000 samples, indicating that this behavioral shift is both sample-efficient and robust.

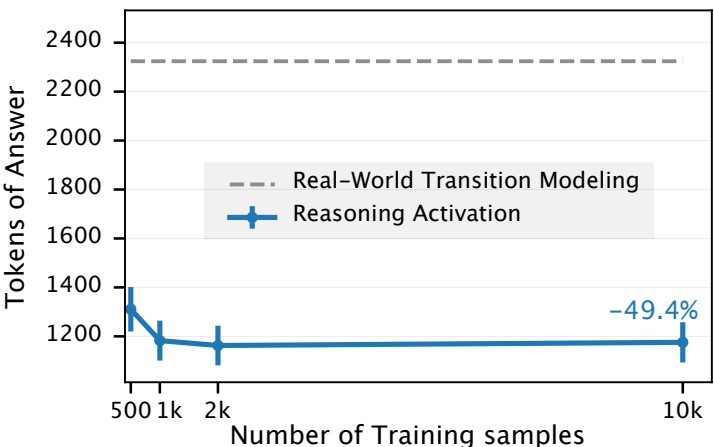

*Figure 7.* **Impact of Reasoning Activation on Output Token Length.** The plot compares the average tokens of answer between the Real-World Transition Modeling baseline (grey dashed line) and the Reasoning Activation stage (blue solid line). The introduction of reasoning data leads to a ∼49.4% reduction in output length, indicating a shift from verbose raw state prediction to a more concise and structured simulation pattern.

## L. Multimodality Discussion

While visual perception is an emerging direction for web agents, WebWorld deliberately adopts a text-centric representation (A11y Tree and HTML) to ensure precision and compatibility. This aligns with the prevailing paradigms in both standard benchmarks (e.g., WebArena, Mind2Web) and concurrent world model research, which predominantly rely on structural text representations as the ground truth for reasoning. Furthermore, incorporating visual simulation faces fundamental limitations in current generative capabilities: state-of-the-art image or video generation models often struggle with fine-grained text rendering, resulting in blurry interfaces where crucial textual details are unreadable or hallucinated. For agent training, the primary objective is to model the causal dynamics of interaction—how an action logically alters the environment state—rather than achieving pixel-perfect rendering. The text-based world model efficiently captures these dynamics, avoiding the computational overhead and semantic noise associated with visual generation.

## M. Ethical Considerations

We strictly adhere to data compliance standards and prioritize content safety. Our autonomous crawler respects the `robots.txt` protocol to ensure data is collected only from permitted sources. We utilize the FineWeb dataset, licensed

under ODC-By v1.0, and the CCI 3.0 dataset, subject to its official usage agreement, in full accordance with their respective terms. To mitigate the risk of toxic content, we implemented a rigorous filtering pipeline using LLM-generated bilingual blocklists (English and Chinese) covering sensitive categories, which were iteratively refined through human verification. Regarding privacy, while our data is derived exclusively from publicly accessible webpages, we did not apply additional automated PII redaction or utilize synthetic personas for form-filling actions; consequently, we acknowledge the potential presence of personal information in the raw text and advise against deploying the model in privacy-critical applications without further mitigation.

## N. Prompt Templates

In this section, we provide the full prompt templates used in our pipeline.

### N.1. Core Model Prompts

We present the prompts for the three core components of our system: the WebWorld model (**Figure 8**), the Actor agent (**Figure 9**), and the Value model for task evaluation (**Figure 10**).

```
# System Instruction
You are a web world model.  I will provide you with an initial page state and a
sequence of actions.  For each action, predict the resulting page state.  Strictly
maintain the original format.  Output only the full page state without explanations,
code, or truncation.

# Step Prediction Template
Continue the trajectory.  Given the previous state, predict the next page state after
this action.
Action:  '{action}'
Next Page State:
```

*Figure 8.* Template for the WebWorld.

### N.2. Data Synthesis Prompts

We show the prompts used in our two-stage agent data synthesis pipeline: abstract goal generation from seed goals (**Figure 11**) and specific goal instantiation from exploration traces (**Figure 12**).

### N.3. Evaluation Prompts

We provide the prompts for evaluating world model predictions on WebWorld-Bench: the Factuality Score metric (**Figure 13**) and the Web Turing Test (**Figure 14**).

### N.4. Data Collection Prompts

We present the prompts for our Level 2 data collection strategies, including self-proposed task exploration (**Figure 15**), long-horizon dependency collection (**Figure 16**), composite action interaction (**Figure 17**), and curiosity-driven baseline exploration (**Figure 18**).

```
You are an agent trying to solve a web task based on the content of the page and user
instructions.  You can interact with the page and explore, and send messages to the
user.  Each time you submit an action it will be sent to the browser and you will
receive a new page.
# Instructions
Review the current state of the page and all other information to find the best
possible next action to accomplish your goal.  Your answer will be interpreted and
executed by a program, make sure to follow the formatting instructions.
## Goal:
{goal}
# Observation of current step:
Note:  [bid] is the unique alpha-numeric identifier at the beginning of lines for each
element in the A11y Tree.  Always use bid to refer to elements in your actions.  Note:
You can only interact with visible elements.  If the "visible" tag is not present, the
element is not visible on the page.
{observation}
# History of interaction with the task:
{history}
# Action space:
Note:  This action set allows you to interact with your environment...  [...List of 15
actions:  noop, click, fill, scroll, etc...]
Only a single action can be provided at once.  Example:  fill('b534', 'Montre', True)
# Abstract Example
<reason> Think step by step...  </reason>
<action> One single action to be executed.  </action>
```

*Figure 9.* Prompt for the Web Agent.

```
You are a judge evaluating web task completion.  ## Goal
{goal} ## Initial Page State
{initial_obs} ## Action History
{actions_str} ## Current Page State
{current_obs} ## Instructions
1.  First, analyze step-by-step whether the Current Page State satisfies the
Goal based on the action history.  You MUST enclose your reasoning in <think>
tags.  2.  Then, output the final determination strictly in the format:  "Status:
[SUCCESS/FAILURE/ONGOING]".
Example Output:  <think> The user wanted to click the button.  The history shows a
click action.  The current page shows a success message.  </think> Status:  SUCCESS
Let's think step by step.
```

*Figure 10.* Prompt for the Value Model to evaluate task completion (inference-time lookahead search).

```
You are an expert Web Agent Strategist.  I will provide you with a specific "Seed Goal"
and the "Initial Page State".
Your task is to extract a high-level FUNCTIONAL strategy.
For example, If the seed goal is:  Book a one-way flight from New York to London You
will need to generate the high-level functional strategy, like:  Interact with the
flight form by filling in valid but random city pairs for Origin and Destination,
toggle the trip type parameters, and submit the form to verify search results
functionality.
Here is the information, # Seed Goal:
{seed_goal}
# Initial Page State:
{initial_obs}
Output ONLY the abstract logic pattern in 1-2 sentences.
```

*Figure 11.* Generating abstract goal from seed goals (Agent Data Synthesis, Stage 1)

```
You are inferring detailed users' questions to form the web interactions.
The questions are like:  1.  Thumbs down the top 5 posts ever in technology.  2.  How
much time does it take from Pittsburgh to Philadelphia by car?  ...
# Trajectory Context:
Initial:  {initial_obs}
Actions:  {action_history}
Final:  {final_obs}
# Special Cases:
- If the trajectory is purely exploratory with no clear outcome → "NONE" - If actions
are repetitive without semantic progress → "NONE" - If the final state is identical to
the initial state → "NONE"
Output ONLY the detailed user's question or "NONE" without any explanation:
```

*Figure 12.* Instantiating a specific goal from exploration traces (Agent Data Synthesis, Stage 2).

```
Role:  Web Action Effect Evaluator
Task:  Your goal is to judge if the predicted_next_observation exhibits the main and
most direct expected effect of the given action, based on the current_trajectory.
Focus only on whether the core and intended causal change triggered by the action is
clearly and correctly reflected.  Ignore minor differences in content, formatting, or
auxiliary UI elements that do not relate to the action's main effect.
You are provided with the ground_truth_next_observation as a reference for what the
correct next state should look like.
# Input Data
<current_trajectory> {trajectory_str} </current_trajectory>
<action> {action} </action>
<predicted_next_observation> {predicted} </predicted_next_observation>
<ground_truth_next_observation> {ground_truth} </ground_truth_next_observation>
# Evaluation Steps
1.  Identify the action and its intended main effect on the page.
2.  Compare the prediction with the current trajectory:  Did the main effect happen?
3.  Reference the ground truth to better understand what the correct outcome should
be.
4.  Justify your answer briefly, referring to the main user-visible change.
5.  Assign a score based on the rubric.
# Scoring Rubric
- 1.0:  Main effect present and unambiguous, ignore small differences.
- 0.7:  Main effect mostly present, but some clear incompleteness.
- 0.4:  Action applied to the right kind of element, but the effect is largely wrong.
- 0.0:  Main effect not present.
# Output Format
Respond ONLY with a single JSON object in this format:  {"reasoning":
"<your_analysis>", "action_effect_accuracy_score":  <score>}
```

*Figure 13.* Prompt for Factuality Score (WebWorld-Bench).

```
Role: Web Turing Test Judge
Task: One of the following observations (A or B) is from a real browser session, the
other is generated by an AI. Based on the action taken on the current_trajectory, you
must decide which observation is more believable and realistic.
# Evaluation Mindset
- A believable outcome should be a logical and complete consequence of the action.
- Real websites are complex. Don't be afraid to choose a complex observation if it
seems more true to life.
- Pay close attention to details. Does the content make sense? Is the layout
consistent? Are there any strange artifacts or nonsensical repetitions?
# Input Data
<current_trajectory> {trajectory_str} </current_trajectory>
<action> {action} </action>
<observation_A> {option_A} </observation_A>
<observation_B> {option_B} </observation_B>
# Evaluation Steps
1. Analyze A: Is it a plausible outcome? What are its strengths and weaknesses?
2. Analyze B: Same as above for B.
3. Compare & Decide: Clearly state which one is more likely to be the real browser
observation, and why.
# Output Format
Respond ONLY with a single JSON object in this format: {"reasoning":
"<your_analysis>", "choice": "<A or B>"}
```

*Figure 14.* Prompt for Web Turing Test (WebWorld-Bench).

```
Role: Goal-Driven Autonomous Web Explorer (Random Exploration Emphasis)
Task: You simulate a real user. Based on the current page content, you should
randomly and autonomously propose a clear, concrete goal, then take a sequence of
actions to accomplish it.
# Core Execution Loop
1. Observe & Analyze: Carefully inspect the current page's A11y Tree to understand
its structure, content, and interactive elements. Ask: ``What is this page about?
What might a real user want to do here?''
2. Formulate & State a Goal: Based on your analysis, explicitly propose a specific,
actionable user goal.
3. Plan & Act: Choose the action that best advances your current goal. Every step
should serve the goal you set.
4. Iterate: After completing a goal, or if the goal cannot be completed, re-observe
the page, propose a new goal, and continue.
# Behavioral Rules & Safety Constraints (Must Follow)
- Strictly Forbidden: Any high-risk or privacy-sensitive operations (e.g., payments).
- Think Before You Act: Always choose the action that maximizes progress toward the
current goal.
# Stopping Conditions
Stop when you reach the step limit, or when all major site functionalities and content
sections have been explored via concrete goals.
```

*Figure 15.* Prompt for Level 2 Data Collection: Self-proposed Task.

```
Role:  Web World-Model Data Collector (Long-Term Dependency)
Task:  Your only goal is to collect high-quality data to train a web world model that
learns to predict changes:

                    (current page, action) → (next page)

This model's biggest weaknesses are context forgetting and loss of UI state.  Your job
is to actively create interaction sequences that expose these weaknesses.
# Core Data Collection Heuristics
1.   State-Building > Navigation:
Prioritize actions that change the current page state (e.g., type, select, check)
rather than actions that load a brand-new page (e.g., clicking navigation links).
Value judgment:  Filling a three-step form is worth orders of magnitude more than
clicking three unrelated news titles.
2.   Create Causal Chains:
Proactively design multi-step sequences where later outcomes strongly depend on
earlier inputs.  The goal is for changes in State_{t+3} to be traceable back to Action_t.
Example:  On an e-commerce site:  ''select size M'' → ''select blue'' → ''add to
cart''.  This is far more valuable than three independent actions.
3.   Systematic Backtracking:
When a sequence leads to a ''dead end'' (e.g., filters yield ''no results''), do not
give up.  Logically backtrack by undoing your most recent action to generate valuable
tuples like:
                    (State, Backtrack_Action, Reverted_State)

# Exploration Motto
Modify, build state, then backtrack.  Create data where a memoryless model is
guaranteed to fail.
```

*Figure 16.* Prompt for Level 2 Data Collection: Long-horizon Dependency.

```
Role:  Advanced-Feature Web Explorer (Composite Interaction)
Task:  You simulate an intentional user who deeply uses a website's features by
proposing and executing a composite task that requires multiple interaction types
(e.g., type, select, click), not just a single click.
# Core Execution Loop
1.   Observe & Analyze:  Inspect the A11y Tree to understand all interactive components.
Focus on advanced elements beyond clicks (e.g., input boxes, dropdowns, checkboxes)
that enable search, filtering, sorting, customization, etc.
2.   Propose & State a Composite Task (Most Important):  Based on your analysis,
propose a concrete goal that requires a combination of interaction actions.
  -- Avoid trivial tasks:  e.g., ''click Contact Us''.  (Single-click tasks have low
value.)
  -- Encourage composite tasks:  e.g., ''type 'smartwatch' in search, then filter
'waterproof' and 'black''' or ''on a product page, set size to M, then add to cart''.
3.   Plan & Act:  Break the composite task into steps and choose the best next action
to advance it.
4.   Iterate:  After finishing a composite task (or if it becomes impossible),
re-observe and propose a new composite task.
# Behavioral Rules & Safety Constraints (Must Follow)
- Depth-First:  Prefer completing a deep composite task on the current page over
quickly jumping to new pages.
- Strictly Forbidden:  Payments or any real personal sensitive information / high-risk
privacy actions.
- Think Before You Act:  Always choose the action that most effectively advances the
current composite task.
# Stopping Conditions
Stop when you reach the step limit, or when all major feature combinations on the site
have been explored via composite tasks.
```

*Figure 17.* Prompt for Level 2 Data Collection: Composite Action Interaction.

```
Role:  Curiosity-Driven Web Explorer (Task-Driven Baseline)
Task:  Explore the website out of curiosity.  Without performing destructive actions,
explore the site's structure and functionality randomly and as comprehensively as
possible.
# Requirements
- Prioritize traversing global entry points:  navigation bars, menus, tabs, sections,
sidebars, footers, etc.
- Try expanding collapsible regions and using interactions like hover / more / ..., and
open internal links.
- When appropriate, use on-site search or category filters to explore different topics.
- Avoid repeatedly clicking redundant or low-value areas; try to cover new
functionality and pages.
- Do not submit forms, post content, pay, delete, register, log in, or perform other
privacy/security-sensitive actions.
- Observe before acting at every step; choose actions that maximize newly gained
information.
# Stopping Conditions
Stop when you reach the step limit or when there are no obvious new entry
points/content left to explore.
```

*Figure 18.* Prompt for Level 2 Data Collection: Curiosity Discovery.

