# OpenReview forum: "WebWorld: A Large-Scale World Model for Web Agent Training"
_ICML.cc/2026/Conference — ICML 2026 regular_

### Official Review · Reviewer_8s7B · 2026-02-24

**Soundness:** 2
**Presentation:** 3
**Significance:** 2
**Originality:** 2
**Overall Recommendation:** 3
**Confidence:** 4

**Summary:**

This paper creates a large-scale web dataset to train world models for the web. Specifically, the authors 1) crawl URLs from FineWeb and a subset of CCI 3.0; 2) use methods such as random walk and LLM-based agents to create tasks and trajectories; 3) use LLMs to evaluate and select trajectories that successfully completed the tasks; 4) uses rule-based heuristics as well as LLMs to filter for high-quality and safe trajectories for final training. Then, the authors first show that models trained with such data performs well on WebWorldBench (also proposed in this work). Next, the authors show that WebWorld models can be used to synthesize training data / help inference time search algorithms on benchmarks such as MiniWob++ and WebArena.

**Compliance With Llm Reviewing Policy:**

Affirmed.

**Final Justification:**

I maintain my slightly negative rating. I believe the experimental setup is solid and that the created dataset can be useful. However, several core concerns I have are still unresolved.

1. In the context of current web agents, there are already a few large-scale multi-modal datasets such as the one from WebDreamer. This makes me uncertain whether creating another text-based dataset is a step forward in the direction of web agent research. Besides, using LLM as a "world model" to help generate synthetic data is also an already explored idea, such as the two papers mentioned by reviewer ZtyR and also Kimi-K2 [1] (see section 3.1.1).

2. I believe it is hard to claim the proposed data creation pipeline is scalable. As indicated during the discussion, the pipeline heavily relies on LLMs and currently costs 20k USD estimated by the author. Although it is much cheaper than human annotation, this is not scalable as further creating more data (e.g., 10x the size) would simply incur linearly more cost (e.g., 10x the cost). Given the current cost is already non-trivial, it would be hard for many researchers and labs to replicate and extend.

---

References:

[1] Team, Kimi, et al. "Kimi k2: Open agentic intelligence." arXiv preprint arXiv:2507.20534 (2025).

**Key Questions For Authors:**

please see weaknesses.

**Limitations:**

yes

**Strengths And Weaknesses:**

Strengths:
- This work creates a new million scale web world model training dataset as well as a test set to benchmark progress in web world models.
- The authors conducted many empirical experiments (e.g., using WebWorld models to synthesize training data or help inference-time search algorithms) and showed that simulations from WebWorld models can be used to further improve the downstream performance of existing web agents.


Weaknesses:
1. The overall data creation pipeline and empirical findings are highly similar to WebDreamer [1], yet this work did not compare to [1] in Table 1-6. Similar to this work, WebDreamer (1) also curates million-scale web world-modeling data from real URLs, and (2) also demonstrates that the resulting world model improves downstream web agent performance on benchmarks such as WebArena.

2. The dataset generated in this work (Section 3.2) is based on a text-only representation of websites (i.e., the accessibility tree). However, real-world websites are highly visual, and much web content (e.g., shopping websites) are naturally presented as images/icons. Why is accessibility-tree-based data used for training? More generally, I believe VLM-based datasets and approaches (e.g., WebDreamer) seem more natural and better aligned with realistic web environments.

3. The paper claims the dataset curation recipe is "scalable" (L201), but in practice it heavily relies on LLMs to synthesize data. It is hard to claim this is scalable, as creating more trajectories would require a lot of LLMs calls to 1) propose tasks, 2) interact with the websites, 3) evaluates whether the tasks are successfully completed; 4) filter for high-quality/safe trajectories. In effect, scaling the dataset appears to require proportionally more compute due to the linear growth in LLM usage.

4. Section 5 evaluates WebWorld on existing web agent benchmarks such as WebArena, either by (1) synthesizing training data or (2) assisting inference-time search. However, relevant baselines for each setting are not included. For data synthesis during training, methods such as WebEvolver [2] should be compared; for inference-time search, approaches such as WebDreamer [1] should be compared.


---

References

[1] Gu, Yu, et al. "Is your llm secretly a world model of the internet? model-based planning for web agents." arXiv preprint arXiv:2411.06559 (2024).

[2] Fang, Tianqing, et al. "WebEvolver: Enhancing Web Agent Self-Improvement with Co-evolving World Model." Proceedings of the 2025 Conference on Empirical Methods in Natural Language Processing. 2025.

---

> ### Author Rebuttal · Authors · 2026-03-29
>
> We sincerely appreciate the reviewer's detailed and rigorous evaluation. We address each concern below. Before addressing the specific weaknesses, we note that both WebDreamer (Section 5.2) and WebEvolver (Table 1, Section 2)  have already been discussed in our paper.
>
> > **W1:  The paper’s novelty relative to WebDreamer is unclear, given the strong similarity in data construction and empirical findings and the lack of direct comparison in Tables 1–6.**
> >
>
> Thank you for the detailed review. The core difference between WebWorld and WebDreamer lies in their **different design goals** for the world model. WebWorld is designed **as an environment simulator for training data synthesis**. By contrast, WebDreamer **uses the world model for inference-time lookahead search**. At each step, it performs forward simulation over candidate actions, so the world model only generates natural language descriptions, not supporting training data synthesis. More importantly, **training is a one-time cost**, while inference incurs recurring costs. We prefer using the world model to improve agent capability during training rather than relying on search.
>
> In **data construction**, WebDreamer adopts a Web Random Walking strategy and misaligns with user distribution. By contrast, we introduce Autonomous Exploration and Task-Oriented Execution to collect data that more closely resembles real users’ web interactions. In **data scale**, while WebDreamer has 3.1M single-step samples, WebWorld contains 1.06M multi-turn trajectories (avg. 5.4 steps), totaling over 5.7M interaction steps, making our dataset nearly 1.8× larger than WebDreamer.
>
> > **W2: The dataset uses accessibility-tree-based (text-only) representation, while real-world websites are highly visual.**
> >
>
> Due to the word limit, please see our response to Reviewer ZtyR (W2).
>
> > **W3: The pipeline heavily relies on LLM calls at every stage, so the "scalable" characterization needs re-examination.**
> >
>
> By “scalable,” we mean two things. First, the pipeline can increase corpus size at low cost without being bottlenecked by human annotation. Second, the additional data continues to improve model performance as the dataset scales. WebWorld satisfies both criteria. Our pipeline is fully automated without human and has produced 1.06M trajectories, which is 100× larger than prior work. Figure 4 further shows a clear scaling trend: more data consistently improves performance, and the training loss follows a power law with no sign of saturation.
>
> **LLM calls are used only in high-value steps (agent exploration and task synthesis in Level 3), and are a one-off cost during data collection.** We believe this may stem from a misunderstanding of our pipeline. In WebWorld, the pipeline is hierarchical, with Level 1 (43.3%, 293K trajectories) being rule-based with no LLM calls, and tasks synthesized by LLM appearing only in Level 3 (16.1%). The “success evaluation” mentioned by the reviewer is not part of our pipeline. As stated in Section 3.3, quality filtering is entirely rule-based.
>
> > **W4: Data synthesis should be compared with methods like WebEvolver, while inference-time search should be compared with approaches like WebDreamer.**
> >
>
> **Inference-time search — WebDreamer comparison.** We have added experiments that strictly replicate WebDreamer's lookahead on MiniWob: generate 3 candidate actions, apply LLM self-refinement to filter irrelevant candidates, use the world model to produce NL state change descriptions (not A11y Tree), score with GPT-4o using 3-level pointwise scoring (Success/On-track/Failure), and execute the highest-scored action. Results (extending Table 6):
>
> | Setting | World Model | Value Model | Fmt | Score | Alg (k) | Reward |
> | --- | --- | --- | --- | --- | --- | --- |
> | Baseline | GPT-4o | GPT-4o | A11y | Point | BoN (3) | 63.8 |
> | Baseline | GPT-5 | GPT-4o | A11y | Pair | BoN (3) | 64.5 |
> | Ours | WebWorld-8B | GPT-4o | A11y | Point | BoN (3) | 64.8 |
> | Ours | WebWorld-8B | GPT-4o | A11y | Pair | BoN (3) | 65.5 |
> | WebDreamer-setting | WebWorld-8B | GPT-4o | NL | Point (3-level) | BoN (3)+SR | 64.5 |
> | Ours | WebWorld-8B | GPT-5 | A11y | Pair | BoN (3) | 67.5 |
> | WebDreamer-setting | WebWorld-8B | GPT-5 | NL | Point (3-level) | BoN (3)+SR | 66.1 |
>
> The results show that WebDreamer setup still underperforms WebWorld.
>
> **Data synthesis — WebEvolver comparison.** A direct comparison with WebEvolver is not feasible because it does not release its model weights or training data (Table 1), and its co-evolution framework relies on unavailable components such as the co-evolution loop, reward signal, and agent-driven data schedule. Moreover, even if reproduced, the comparison would entangle the effect of world model quality with that of the co-evolution strategy, rather than isolating the contribution of world model data as in WMA and WebSynthesis (Appendix C). We therefore treat WebEvolver as related but not directly comparable.

---

> > ### Author Rebuttal · Reviewer_8s7B · 2026-04-02
> >
> > Thank you for your response to W1 and W4. I believe they mostly addressed my concern. I encourage the authors to include these clarifications in the revision, as prior work such as WebDreamer remains highly relevant.
> >
> > > W2: The dataset uses accessibility-tree-based (text-only) representation...
> >
> > I agree that current VLMs tend to achieve stronger performance using text representations. However, I do not believe this justifies focusing dataset construction on text rather than visual modality. Rather, these results highlight the need to improve visual-based training, as tasks such as web are inherently visual. I stand by my position that large scale text-based dataset for web is highly limited, especially when there are already prior work such as WebDreamer which constructs similar (but for different purpose) data using images.
> >
> >
> > > W3: The pipeline heavily relies on LLM calls at every stage, so the "scalable" characterization needs re-examination.
> >
> > "First, the pipeline can increase corpus size at low cost without being bottlenecked by human annotation." Could you provide a quantitatively cost breakdown? If large-scale data generation requires extensive LLM calls - potentially with strong models for quality control - the overall cost may be significant.
> >
> > "The 'success evaluation' mentioned by the reviewer is not part of our pipeline" On L209-210, you claim "Agents then execute these tasks on the corresponding websites, and we retain only successful trajectories". Perhaps I misunderstood? Additionally, since these "tasks" are proposed by an LLM as well (L201-203), I assume to determine whether the agent "successfully" solved the task also relies on LLM-as-a-judge.
> >
> > "LLM calls are used only in high-value steps (agent exploration and task synthesis in Level 3)" Beyond exploration and task synthesis in level 3, In Section 3.3 Filtering and Section 3.5 CoT Synthesis the paper also stated LLMs are used?

---

> > > ### Author Response · Authors · 2026-04-04
> > >
> > > > W2: Web tasks are inherently visual, so the world model should ideally be vision-based.
> > > >
> > >
> > > We greatly appreciate this concern. We also believe the development of web world models should naturally progress from text to images to video, increasingly close to the real world.
> > >
> > > In fact, we also note that image web simulation is a natural next step, and we have already initiated this line of research. As current image generation models remain insufficient for web simulation, we still adopt a text-centric approach even for visual simulation: we follow a text-to-rendered-image pipeline built upon a fully rewritten GUI rendering engine, and plan to release this as a subsequent publication.
> > >
> > > Returning to this work, we think a research contribution requires all the necessary conditions to be in place. The key question is therefore whether the capabilities of text, image, and video generation models are sufficient to support web world modeling in their respective modalities. Currently, text generation models are well-established for web simulation, offering the best ROI among all modalities. On the image and video side, we have already extensively evaluated Veo3, GPT-Image, and Nano Banana Pro. The generated text in web screenshots is completely blurry, and cross-step consistency breaks down. For example, a simulated webpage on a retail site may spuriously jump from Walmart to Amazon.
> > >
> > > WebDreamer is an excellent work that pointed toward the direction of visual web world models, though it outputs natural language descriptions rather than visual images. From its release in 2024 to today in 2026, image and video generation models for high-information-density domains still remain technically immature. Nevertheless, we firmly believe that web world models to be inherently multimodal as the underlying generative capabilities mature.
> > >
> > > > W3: The pipeline heavily relies on LLM calls at every stage, so the "scalable" characterization needs re-examination.
> > > >
> > >
> > > We thank the reviewer for the thorough reading. For clarity, we provide a complete cost breakdown covering all LLM-dependent steps in the pipeline. We estimate costs using GPT-5 pricing ($1.25/1M input, $10.00/1M output), except Stage 2 CoT which uses Claude-Opus-4.1 ($15/1M input, $75/1M output) as stated in the paper. *(Note: our previous response incorrectly stated "The 'success evaluation' mentioned by the reviewer is not part of our pipeline"; please refer to the paper and the cost table below, which are both correct.)*
> > >
> > > | Pipeline Step | # Calls | Avg Tokens (In/Out) | Est. Cost (USD) |
> > > | --- | --- | --- | --- |
> > > | Tier 2: Goal Generation | 38K | 3,000 / 200 | $72 |
> > > | Tier 2: Agent Steps | 575K | 4,000 / 300 | $4,623 |
> > > | Tier 3: Task Synthesis | 160K | 800 / 400 | $800 |
> > > | Tier 3: Agent Execution | 752K | 4,000 / 300 | $6,016 |
> > > | Tier 3: Success Evaluation | 150K | 5,000 / 300 | $1,388 |
> > > | URL Quality Scoring | 68K | 2,000 / 300 | $374 |
> > > | Data Enrichment | 390K | 2,500 / 1,200 | $5,913 |
> > > | CoT Synthesis | 1K | 15,000 / 8,000 | $825 |
> > > | **Total API Cost** |  |  | **~$20,011** |
> > >
> > > Comparison with Human Annotation:
> > >
> > > | Pipeline Step | # Items | Time/Item | Total Hours | Cost @$12/hr | Cost @$25/hr |
> > > | --- | --- | --- | --- | --- | --- |
> > > | Trajectory Recording (Tier 2+3) | 226K | 20 min | 75,333 hrs | $904,000 | $1,883,333 |
> > > | Task Writing + Variants + Paraphrase | 160K | 3 min | 8,000 hrs | $96,000 | $200,000 |
> > > | Success Evaluation | 150K | 3 min | 7,500 hrs | $90,000 | $187,500 |
> > > | URL Quality Assessment | 68K | 2 min | 2,267 hrs | $27,200 | $56,667 |
> > > | Data Enrichment  | 390K | 8 min | 52,000 hrs | $624,000 | $1,300,000 |
> > > | CoT Reasoning Annotation | 1K | 30 min | 500 hrs | $6,000 | $12,500 |
> > > | **Total** |  |  | **145,600 hrs** | **$1,747,200** | **$3,640,000** |
> > >
> > > Summary:
> > >
> > > | Method | Total Cost | Time | vs. LLM |
> > > | --- | --- | --- | --- |
> > > | **LLM API** | **~$20,011** | 1 week | **1×** |
> > > | Human @$12/hr (crowdsource) | ~$1,747,200 | ~145,600 hrs | **87×** more expensive |
> > > | Human @$25/hr (US professional) | ~$3,640,000 | ~145,600 hrs | **182×** more expensive |
> > >
> > > **Scalability in data curation for LLM training is primarily determined by the degree of human involvement, not API cost.** Our entire pipeline costs approximately USD 20,011, while equivalent human annotation would require approximately USD 1.7M and 73 person-years. For further context, curating data for a single capability in a frontier model (e.g., search grounding, where API costs alone can reach approximately USD 1M) already exceeds our total pipeline cost by orders of magnitude.
> > >
> > > **Price References:**
> > >
> > > - Price of GPT-5: https://developers.openai.com/api/docs/pricing
> > > - Price of Claude-Opus-4.1: https://platform.claude.com/docs/en/about-claude/pricing
> > > - US annotation avg: https://www.ziprecruiter.com/Salaries/Data-Annotation-Salary
> > > - Annotation platform rates: https://www.dataannotation.tech/faqs
> > > - Offshore annotation: https://www.gdsonline.tech/data-annotation-pricing/

---

### Official Review · Reviewer_ZtyR · 2026-03-09

**Soundness:** 3
**Presentation:** 3
**Significance:** 4
**Originality:** 3
**Overall Recommendation:** 5
**Confidence:** 3

**Summary:**

This paper introduces WebWorld, a large-scale web world model trained on 1.06M real-world interaction trajectories from the open web. The model is formulated as next-state simulator over instruction and interaction history, and is trained with a two-stage recipe: large-scale transition modeling followed by lightweight reasoning activation using 1K CoT samples. The paper also proposes WebWorld-Bench, an intrinsic benchmark with two judge-based metrics across nine dimensions, and reports strong extrinsic gains when WebWorld-synthesized trajectories are used to fine-tune downstream agents. Overall, the paper aims to establish a scalable recipe for building web simulators that are useful not only intrinsically, but also for actual agent training.

**Compliance With Llm Reviewing Policy:**

Affirmed.

**Key Questions For Authors:**

Although enhancing Web Agent with world model is not new, this paper is engineering-wise solid and could be impactful for the community. It would be better if authors can discuss more recent work in this direction, such as:

- Agent Learning via Early Experience (https://arxiv.org/pdf/2510.08558)
- Dyna-Mind: Learning to Simulate from Experience for Better AI Agents (https://arxiv.org/abs/2510.09577)

**Limitations:**

yes

**Strengths And Weaknesses:**

Strengths

1. The paper addresses a central bottleneck in web-agent research: collecting real-world trajectories is slow, rate-limited, and potentially risky, while prior web world models are mostly limited to closed benchmarks and relatively small datasets. This makes the work timely and practically relevant. If the dataset can be open-sourced, it could become a highly valuable resource for the community.

2. The three-level data collection strategy is well designed and substantially broadens coverage compared to prior work.

3. The combination of intrinsic and extrinsic evaluation is convincing.

Weaknesses

1. The benchmark is generated using the same hierarchical data curation pipeline as the training set, albeit held out. It raises a concern that WebWorld-Bench may be aligned with the authors’ training distribution and modeling assumptions, which could overstate generalization

2. The paper supports multiple textual formats, but the overall representation remains text-only. This is a design choice with clear trade-offs. This is reasonable for the current paper, but it also limits applicability to settings where visual fidelity and layout information matter. This direction could be left for future work.

---

> ### Author Rebuttal · Authors · 2026-03-29
>
> We are grateful for the reviewer's thoughtful evaluation and constructive suggestions.
>
> > **W1: Distribution alignment between WebWorld-Bench and training data**
>
> We have conducted a quantitative distribution analysis between the training set and WebWorld-Bench. The URL exact-match rate between the two sets is **0%**, confirming zero sample-level leakage. The domain overlap rate is approximately **32%**, substantially lower than within-training random splits (**~91%**). Overlapping domains (e.g., both sets containing pages from the medical domain) reflect the natural distribution of the open web.
>
> If the model merely memorized the training data, it would fail to generalize to environments with no overlap with the web pipeline. However, Table 8 demonstrates that WebWorld transfers effectively to code, game, GUI, and API environments. The four domains entirely outside the web curation pipeline, achieving an average +22% Total Score improvement. Table 5 further shows that agents trained on WebWorld-synthesized data achieve significant gains on MiniWob++ (+9.9%) and WebArena (+10.9%), both community-standard benchmarks fully independent of our pipeline. Additionally, proprietary models (GPT-4o, Claude-Opus-4.1, Gemini-3-Pro) that have never been exposed to our pipeline produce well-differentiated scores on WebWorld-Bench (Table 3).
>
> > **W2: The text-only representation of web environments limits its applicability to visual information.**
>
> From the experimental results, the text-only approach performs better than the visual-based one. WebDreamer’s GPT-4o VLM prompting improves WebArena performance from 17.6% to 23.6% (+6.0%), while Qwen3-14B + WebWorld reaches 24.3%, a larger gain of 9.2%, and comes close to GPT-4o’s 26.6%.
>
> Current visual web world model approaches, such as WebDreamer, still produce **text rather than images**. A true visual web world model (screenshot → screenshot) is highly challenging to scale, because current image generation models cannot reliably render fine-grained web UI, often suffering from blurry text, layout errors, and missing elements. Under the technical constraints, WebWorld follows the **text-only design** adopted by all existing trained web world models (WMA, DreamGym, WebSynthesis, Word2World, and WebEvolver; Table 1). If one insists on using visual inputs, but the model cannot produce visual outputs, the only fallback is natural-language state descriptions. This input–output mismatch makes the resulting model unsuitable for large-scale synthesis of agent training trajectories, leaving inference-time lookahead search as the only practical use case. However, in today’s regime where inference compute is often more valuable than training compute, this trade-off is not cost-effective.
>
> Multimodal is a natural direction for future work. As image generation technology advances, text + visual fusion in world models is a promising avenue to explore.
>
> > **Additional References**
>
> We thank the reviewer for recommending Early Experience and Dyna-Mind. Both works explore injecting environment knowledge into the agent itself (implicit/internal world model). Early Experience lets agents propose alternative actions at each state in expert trajectories to implicitly learn environment dynamics; Dyna-Mind embeds simulation into the agent's chain-of-thought reasoning, teaching agents to predict alternative futures during inference.
>
> The core distinction from WebWorld is that both approaches inject world modeling inside the agent policy and **rely on online interaction with real environments** for data collection. WebWorld trains a **standalone external world model** that, once trained, operates entirely offline. It can synthesize training trajectories for any downstream agent without further environment interaction. The two approaches are complementary: WebWorld provides a simulation environment, while Early Experience and Dyna-Mind enhance agents' internal understanding of the environment. We will discuss both works in the revised Related Work section.

---

> > ### Author Rebuttal · Reviewer_ZtyR · 2026-03-31
> >
> > Authors' response has solved my problem.

---

### Official Review · Reviewer_1RZh · 2026-03-13

**Soundness:** 3
**Presentation:** 3
**Significance:** 4
**Originality:** 4
**Overall Recommendation:** 5
**Confidence:** 3

**Summary:**

The paper introduces WebWorld, an open-web world-model family (8B, 14B, and 32B variants) for training web agents. The paper also introduces WebWorld-Bench, an intrinsic evaluation suite with Factuality Score and Web Turing Score across nine dimensions, and demonstrates that 8,000 WebWorld-synthesized trajectories can improve the performance of Qwen3 on MiniWob++ and WebArena. Additional analyses include inference-time search, scaling behavior, reasoning ablations, and cross-environment generalization to API services, code, GUI, and game domains.

**Compliance With Llm Reviewing Policy:**

Affirmed.

**Final Justification:**

The authors' rebuttal addressed my concerns. The extra manual calibration data and information provided have convinced me that the proposed pipeline is a feasible and practical solution. Therefore, I maintain my Accept recommendation.

**Key Questions For Authors:**

1. Is there any human calibration for evaluation on a representative subset? Strong agreement can help increase confidence in the intrinsic benchmark.
2. Although the benchmark is held out, it is generated by the same pipeline. Is there disjointness at the URL, domain, site-family, or template level?

**Limitations:**

yes

**Strengths And Weaknesses:**

Strengths:
1. The presentation is clear: the paper is readable, the narrative is easy to follow, and the appendix is unusually rich for a systems paper, covering training details, action space, and prompt templates.
2. The paper addresses an important bottleneck that real web interaction is slow and constrained, so an open-web simulator for agent training could be valuable infrastructure. The paper also shows downstream gains, scaling behavior, and transfer beyond a single benchmark.

Weaknesses:
1. The intrinsic evaluation appears to be primarily LLM-judge based, without human-calibration evidence. This weakens confidence in the reliability of the intrinsic benchmark.

---

> ### Author Rebuttal · Authors · 2026-03-29
>
> We sincerely appreciate the reviewer's careful evaluation and positive assessment of our work.
>
> > **W1: Is there any human calibration for evaluation on a representative subset? Strong agreement can help increase confidence in the intrinsic benchmark.**
>
> We have conducted a human evaluation to verify the reliability of WebWorld-Bench. We randomly sampled ~50 samples from WebWorld-Bench and had eight annotators from diverse backgrounds, including PhD students, Master's students, and industry professionals across four different disciplines, independently score them. To ensure consistency, we provided all annotators with a detailed scoring guideline that defines the evaluation criteria and illustrates each score level with examples. We then compared the human judge scores against GPT-4o and Claude-Opus-4.1 on the same sample set (extending Table 4):
>
> | Model              | GPT-4o Fact. | GPT-4o Tur. | Claude-Opus-4.1 Fact. | Claude-Opus-4.1 Tur. | Human Fact. | Human Tur. |
> |--------------------|--------------|-------------|-----------------------|----------------------|-------------|------------|
> | GPT-4o             | 59.5         | 35.4        | 51.6                  | 21.0                 | 65.6        | 37.2       |
> | Claude-Sonnet-4.5  | 59.7         | 36.2        | 59.9                  | 31.9                 | 63.4        | 39.6       |
> | Gemini-3-Pro       | 70.3         | 43.2        | 72.8                  | 36.5                 | 73.5        | 39.0       |
> | Qwen3-14B          | 38.0         | 22.7        | 35.3                  | 15.8                 | 37.1        | 19.8       |
> | WebWorld-8B        | 70.1         | 42.2        | 67.6                  | 31.7                 | 70.6        | 41.1       |
>
> In terms of absolute scores, human judges and the GPT-4o judge typically differ by fewer than 5 points on both metrics; the Claude-Opus-4.1 judge shows larger deviations on individual models but preserves the same overall trends. Aggregating across both Factuality and Turing (n=10), the Spearman correlation between human and GPT-4o judge is ρ = 0.90 (p < 0.001), and between human and Claude-Opus-4.1 judge is ρ = 0.84 (p < 0.01). On Factuality, all three judges produce an identical model ranking; on Turing, minor rank swaps occur among closely scored models, but the top-tier and bottom-tier groups remain consistent. Since WebWorld-Bench is designed to discriminate between models rather than assign absolute scores, this strong rank correlation confirms its reliability.
>
> > **W2: Although the benchmark is held out, it is generated by the same pipeline. Is there disjointness at the URL, domain, site-family, or template level?**
>
> We have conducted a quantitative distribution analysis between the training set and WebWorld-Bench. The URL exact-match rate between the two sets is **0%**, confirming zero sample-level leakage. The domain overlap rate is approximately **32%**, substantially lower than within-training random splits (**~91%**). Overlapping domains (e.g., both sets containing pages from the medical domain) reflect the natural distribution of the open web.
>
> If the model merely memorized the training data, it would fail to generalize to environments with no overlap with the web pipeline. However, Table 8 demonstrates that WebWorld transfers effectively to code, game, GUI, and API environments. The four domains entirely outside the web curation pipeline, achieving an average +22% Total Score improvement. Table 5 further shows that agents trained on WebWorld-synthesized data achieve significant gains on MiniWob++ (+9.9%) and WebArena (+10.9%), both community-standard benchmarks fully independent of our pipeline. Additionally, proprietary models (GPT-4o, Claude-Opus-4.1, Gemini-3-Pro) that have never been exposed to our pipeline produce well-differentiated scores on WebWorld-Bench (Table 3).

---

> > ### Author Rebuttal · Reviewer_1RZh · 2026-04-04
> >
> > Thanks for the human evaluation to verify the reliability of WebWorld-Bench, which helps increase the confidence in the intrinsic benchmark. The 0% URL overlap and 32% domain overlap demonstrate disjointness. This address my concern.

---

### Decision · Program_Chairs · 2026-04-30

**Decision:**

Accept (regular)

**Comment:**

WebWorld presents a large-scale, text-based web world model and associated benchmark (WebWorld-Bench) built from 1.06M real interaction trajectories, plus extrinsic results showing that WebWorld-synthesized trajectories improve several downstream agents on MiniWob++, WebArena, and non-web domains (code, GUI, games, APIs). Reviewers are aligned that the paper is clearly written, methodologically careful, and empirically substantial: the three-tier data collection and filtering pipeline is documented in detail, the intrinsic benchmark is human-calibrated and shown to correlate well with human judgments, and the extrinsic gains and scaling analyses indicate that WebWorld is genuinely useful as infrastructure. The authors also provide quantitative evidence of train/benchmark disjointness (0% URL overlap, moderate domain overlap) and show that world-model training does not catastrophically harm, and sometimes improves, general multimodal capabilities.

The main disagreement centers on scope and novelty. Reviewer 8s7B argues that, given existing multimodal web world-model work (e.g., WebDreamer) and large web-scale datasets, another text-only dataset is an incremental step and that the pipeline’s ~\$20k API cost does not meet their bar for “scalable.” Other reviewers accept text-only as an engineering choice given current limitations of visual web simulators, and interpret “scalable” as removal of human-annotation bottlenecks plus monotonic performance gains as data increases—criteria that WebWorld convincingly meets. They also note that the pipeline and resulting open resources are likely to be impactful for both industry and research, even if visual world modeling remains an important next step. Taking into account the thorough rebuttal, human calibration, additional comparisons, and the strong practical value of an open, well-documented web simulator and benchmark, the balance of evidence favors **acceptance**, with the expectation that the final version will (i) more explicitly position WebWorld relative to visual and multimodal approaches, and (ii) temper claims about “scalability” to reflect its linear API-cost growth.